# Ubiquitin ligase CHFR mediated degradation of VE-cadherin through ubiquitylation disrupts endothelial adherens junctions

Chinnaswamy Tiruppathi [1] ✉, Dong-Mei Wang[1], Mohammad Owais Ansari [1], Shabana Bano[1], Yoshikazu Tsukasaki [1], Amitabha Mukhopadhyay[1], Jagdish C. Joshi[1], Christian Loch[2], Hans W. M. Niessen[3] & Asrar B. Malik [1] ✉

Vascular endothelial cadherin (VE-cadherin) expressed at endothelial adherens junctions (AJs) is vital for vascular integrity and endothelial homeostasis. Here we identify the requirement of the ubiquitin E3-ligase CHFR as a key mechanism of ubiquitylation-dependent degradation of VE-cadherin. CHFR was essential for disrupting the endothelium through control of the VE-cadherin protein expression at AJs. We observe augmented expression of VE-cadherin in endothelial cell (EC)-restricted *Chfr* knockout (*Chfr^ΔEC*) mice. We also observe abrogation of LPS-induced degradation of VE-cadherin in *Chfr^ΔEC* mice, suggesting the pathophysiological relevance of CHFR in regulating the endothelial junctional barrier in inflammation. Lung endothelial barrier breakdown, inflammatory neutrophil extravasation, and mortality induced by LPS were all suppressed in *Chfr^ΔEC* mice. We find that the transcription factor FoxO1 is a key upstream regulator of CHFR expression. These findings demonstrate the requisite role of the endothelial cell-expressed E3-ligase CHFR in regulating the expression of VE-cadherin, and thereby endothelial junctional barrier integrity.

The central pathogenic feature of degradation of VE-cadherin (a.k.a. CDH5), located at endothelial adherens junctions (AJs), is the breakdown of endothelial junctional barrier leading to unchecked leakage of plasma proteins into tissue and the influx of blood cells[1–7]. VE-cadherin forms $Ca^{2+}$-dependent homophilic *cis* and *trans* dimers at AJs that establish the restrictive nature of endothelial cell-to-cell adhesions preventing free passage of plasma and blood constituents[1,2,8–10]. The cytoplasmic domain of VE-cadherin interacts with β-catenin, plakoglobin, and p120-catenin, and transfer signals into cells though interaction with the actin cytoskeleton[8–10]. Phosphorylation of VE-cadherin at Y-658, Y-685, and Y-731 by *Src* tyrosine kinase was shown to disrupt endothelial AJ integrity[11–16]. Inflammatory mediators as diverse as thrombin, bradykinin, histamine, VEGF, and reactive oxygen species increase vascular endothelial permeability through phosphorylation-

dependent internalization of VE-cadherin resulting in disassembly of VE-cadherin junctions and internalization[11–16]. Tyrosine phosphatases also regulate junctional integrity in contrast through dephosphorylation of VE-cadherin[17]. The key endothelial cell-specific protein tyrosine phosphatase (VE-PTP) interacts with VE-cadherin at AJs where both proteins contribute to endothelial barrier stability[16–20]. Diseases such as acute lung injury (ALI) and its severe form, acute respiratory distress syndrome (ARDS) induced by pneumonia, sepsis, and viral infections such as influenza and SARS CoV2 (COVID-19) also induce endothelial barrier breakdown secondary to disruption or loss of VE-cadherin expression at AJs[3–7,21,22].

The key question that remains incompletely addressed is what regulates the protein expression of VE-cadherin at AJs. Protein ubiquitylation, a well described post-translational modification of

[1]Department of Pharmacology and Regenerative Medicine and The Center of Lung and Vascular Biology, University of Illinois College of Medicine, Chicago, IL, USA. [2]AVMBioMed, King of Prussia, PA, USA. [3]Department of Pathology, VU University Medical Center, Amsterdam, the Netherlands. ✉e-mail: tiruc@uic.edu; abmalik@uic.edu

proteins essential for proteasome-mediated degradation of proteins[23–25], is key to addressing this question. Ubiquitin is a 76-amino acid-containing protein expressed in mammalian cells[23]. Ubiquitylation requires the action of 3 different enzymes (E1, E2, and E3)[24,25]. The final step in the process is covalently linking of ubiquitin (Ub) to the target protein by a E3 ligase[23–25]. The most common forms of ubiquitylation involve the formation of $K^{48}$-linked poly-Ub chains that induce protein degradation through activation of the proteasome pathway[23–25] and the formation of $K^{63}$-linked poly-Ub chain important for protein signaling and trafficking[23–25]. E3 ligases are specific in mediating the ubiquitylation of target proteins. The human genome encodes ~600 E3 ligases; however, whether specific E3 ligase mediate ubiquitylation and degradation of VE-cadherin during vascular inflammation is unknown.

Here we show that the E3 ligase CHFR (*checkpoint protein with FHA and Ring domain*) as a requisite for $K^{48}$-linked poly-ubiquitylation of VE-cadherin and subsequent degradation. Depletion of CHFR in human lung microvascular endothelial cells in culture or EC-specific deletion of *Chfr* in mice (*Chfr$^{\Delta EC}$*) prevented VE-cadherin ubiquitylation and proteasomal degradation. Results show the essential role of CHFR in regulating the integrity of AJs in the basal state and in response to inflammatory mediators.

## Results

### E3 ligase CHFR binds and ubiquitylates VE-cadherin

We used AVM BIOMED Snapshot Proteomics Microarray system to identify VE-cadherin interacting proteins as shown in the workflow (Fig. 1a). The microarray contains 21,065 human recombinant proteins immobilized onto nitrocellulose membrane grids. We identified the binding of soluble C-terminal green fluorescent protein (GFP)-tagged *h*VE-cadherin to proteins in the microarray (Fig. 1a). CHFR showed the strongest interaction with VE-cadherin as defined by the mean binding signal compared to other proteins (Fig. 1b). CHFR was the most prominent E3 ligase interacting with VE-cadherin, (Fig. 1c).

CHFR contains the N-terminal forkhead-associated (FHA) domain interacting with phosphoproteins, Ring finger (RF) domain having E3-ubiquitin ligase, cysteine-rich domain (CR) responsible for protein binding, and a poly-ADP ribose binding zinc-finger (PBZ) motif in the C-terminus binds to poly ADP-ribose polymerases, which is required for the checkpoint function of CHFR[26–28]. To determine interactions between CHFR and VE-cadherin, we generated N-terminal GFP fused wildtype CHFR (WT-CHFR) and CHFR mutants (Fig. 1d) and transfected these into the endothelial cell line (HMEC). CHFR lacking FHA (ΔFHA-CHFR) or PBZ (ΔPBZ-CHFR) domain showed markedly reduced expression (Fig. 1e) whereas CHFR lacking RF domain (ΔRF-CHFR) showed increased CHFR expression (Fig. 1e), indicating the key role of RF domain in mediating autoubiquitylation and degradation of CHFR[29]. Next, we used the anti-GFP agarose bead pull-down assay to assess CHFR binding to VE-cadherin. Here we observed significantly reduced binding of VE-cadherin with the CHFR mutant lacking CR (ΔCR-CHFR) or RF (ΔRF-CHFR) domain (Fig. 1f). Interestingly, CHFR lacking FHA domain (ΔFHA-CHFR) showed 2-fold greater increase in binding to VE-cadherin than WT-CHFR, whereas CHFR lacking PBZ (ΔPBZ-CHFR) domain showed several-fold greater increase in binding to VE-cadherin than WT-CHFR (Fig. 1f).

Studies were next carried out to determine the role of CHFR in ubiquitylating VE-cadherin. We used HEK293 cells (which do not express VE-cadherin) transfected with HA-tagged ubiquitin (HA-Ub) alone or with CHFR-expressing plasmids, and then infected the cells with adenovirus expressing C-terminal GFP-tagged mouse VE-cadherin (Fig. 1g, h). $K^{48}$-linked but not $K^{63}$-linked ubiquitylation of VE-cadherin was seen in the cells expressing HA-Ub, VE-cadherin, and WT-CHFR together (Fig. 1h). We did not observe $K^{48}$-linked ubiquitylation of VE-cadherin in cells expressing VE-cadherin alone or HA-Ub alone (Fig. 1h). Since the RF domain is essential for CHFR ligase function[26] and FHA

domain of CHFR interacts with phosphorylated substrates[27], we next determined whether these two CHFR mutants ubiquitylated VE-cadherin. Defective ubiquitylation of VE-cadherin was seen in cells expressing either ΔRF-CHFR or ΔFHA-CHFR with HA-Ub and VE-cadherin (Fig. 1h), indicating the essential role of both CHFR RF and FHA domains in mediating VE-cadherin binding and ubiquitylation.

### CHFR-mediated VE-cadherin degradation via ubiquitylation induces endothelial barrier breakdown

We next investigated the role of CHFR-mediated VE-cadherin ubiquitylation in promoting VE-cadherin degradation and subsequently the breakdown of AJs. Here we used primary human lung microvascular endothelial cells (HLMVEC) in which CHFR expression was shown to be suppressed using siRNA in a concentration-dependent manner (Fig. 2a). Results also showed time-dependent decreases in VE-cadherin expression in response to LPS in control HLMVEC and recovery thereafter occurring at 24 h, whereas LPS had no effect on VE-cadherin expression in CHFR-depleted HLMVEC (Fig. 2b) consistent with the key role of CHFR in ubiquitylating VE-cadherin.

We did not detect low molecular weight VE-cadherin protein bands using antibodies targeting the VE-cadherin N-terminus or C-terminus for immunoblotting analysis to assess degradation of VE-cadherin (Supplementary Fig. 1a). Interestingly, CHFR depletion prevented LPS-induced degradation of the VE-cadherin-interacting proteins VE-PTP and β-catenin (Supplementary Fig. 1b), whereas LPS failed to downregulate another VE-cadherin interacting protein p120-catenin in either control or CHFR depleted cells (Supplementary Fig. 1b). In a control experiment, to ensure the specificity of CHFR in degrading VE-cadherin, we also used another CHFR-siRNA construct (CHFR-siRNA2) to deplete CHFR and observed that CHFR depletion prevented LPS-induced degradation of VE-cadherin in HLMVEC (Supplementary Fig. 1c).

We next determined the role of CHFR-mediated VE-cadherin ubiquitylation via $K^{48}$-linked poly-Ub chains induced the breakdown of endothelial AJs. CHFR depletion prevented both baseline and LPS-induced VE-cadherin ubiquitylation via $K^{48}$-poly-Ub chains (Fig. 2c). We also determined ubiquitylation of VE-cadherin via $K^{63}$-linked poly-Ub chains in control and CHFR- knockdown HLMVEC, and in contrast, failed see VE-cadherin ubiquitylation via $K^{63}$-linked poly-Ub chains in control and CHFR knockdown HLMVEC with or without LPS challenge (Fig. 2c).

CHFR depletion also prevented LPS-induced downregulation of VE-cadherin expression at AJs (Fig. 2d). Furthermore, we observed that CHFR depletion prevented LPS- and protease-activated receptor 1 (PAR-1)-induced increases in endothelial permeability (Fig. 2e, f). Next, on determining expression by immunostaining, we observed that CHFR expression in the cytosol, nucleus, and at AJs co-localized with VE-cadherin in HLMVEC (Fig. 2g). CHFR was also found in endocytosed vesicles proximal to AJs where it co-localized with VE-cadherin after LPS challenge (known to disrupt VE-cadherin junctions) (Fig. 2g).

### *Chfr* deletion in mice prevents VE-cadherin degradation in vivo

To study the in vivo role of the EC-expressed *Chfr*, we engineered endothelial cell (EC)-restricted *Chfr* knockout *(Chfr$^{\Delta EC}$)* mice. A targeting vector was prepared to flank exon 3 of mouse *Chfr* with 2 loxP sites. *Chfr$^{fl/fl}$* mice were crossed with B6.Cg-Tg(Cdh5-Cre)7Mlia/J (VE-cad-Cre) mice to derive *Chfr$^{\Delta EC}$* mice (Supplementary Fig. 2a, b). We did not observe any developmental defects in *Chfr$^{\Delta EC}$* mice. For the studies described below, we used 6-8wk old male and female *Chfr$^{fl/fl}$* (WT) and *Chfr$^{\Delta EC}$* mice to determine the role of CHFR in regulating endothelial permeability in vivo.

Loss of CHFR protein expression was evident in lung EC of *Chfr$^{\Delta EC}$* mice (Fig. 3a). Using lung tissue to determine VE-cadherin expression in lung EC, we observed greater than 3-fold increase in *Chfr$^{\Delta EC}$* mice as compared to WT (Fig. 3b, c). We also noted augmented expression of

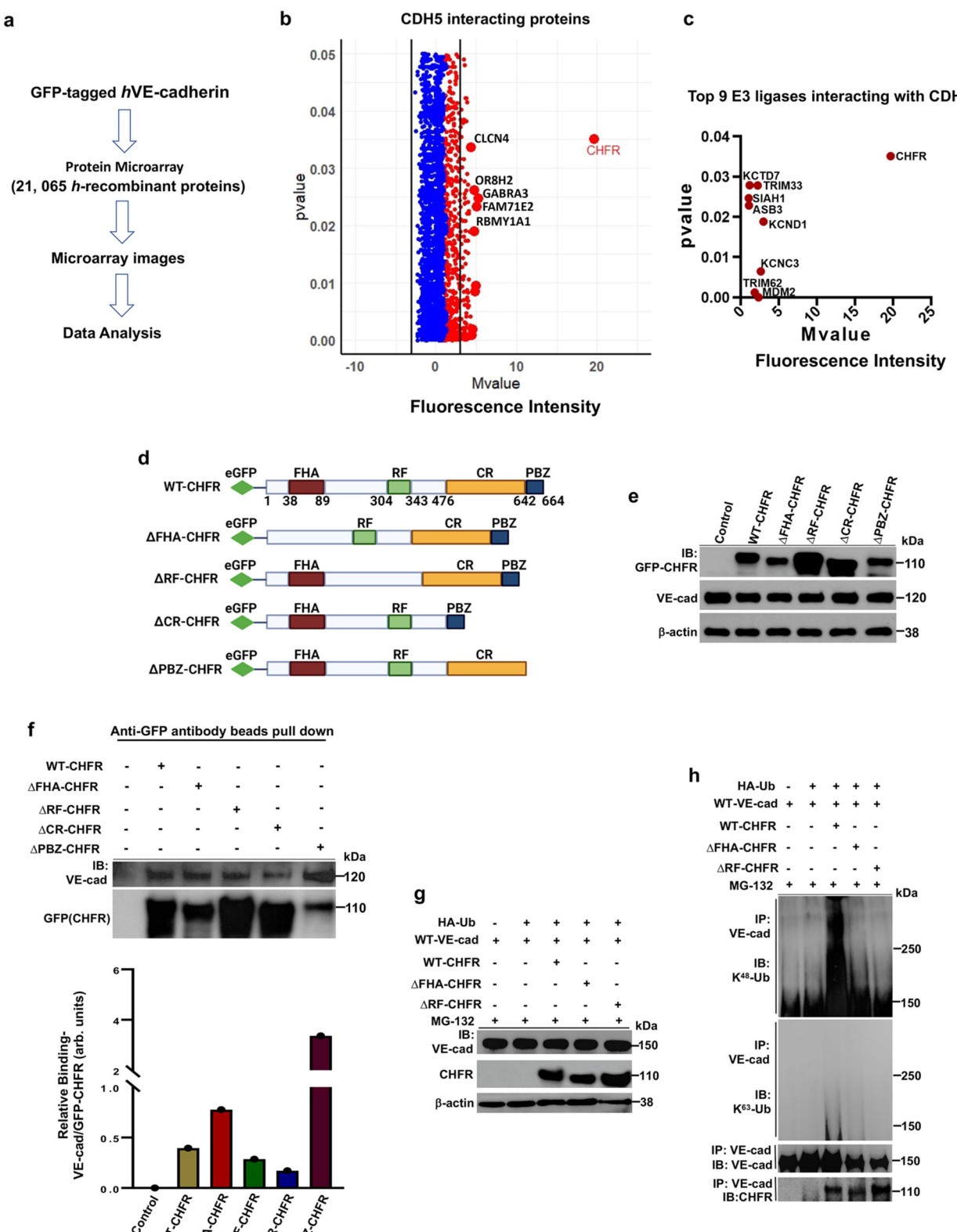

the VE-cadherin interacting proteins VE-PTP (Fig. 3d), p120-catenin and β-catenin (Fig. 3e), and tight junction protein claudin-5 (Fig. 3e) in $Chfr^{\Delta EC}$ mice. These results show that endothelial-specific deletion of CHFR in mice augmented VE-cadherin expression at AJs and recruited the cognate binding partners p120-catenin, β-catenin, and VE-PTP to enhance integrity of AJs.

We next investigated whether LPS influenced the expression of VE-cadherin in $Chfr^{\Delta EC}$ as compared to $Chfr^{fl/fl}$ (WT) mice. LPS (10 mg/kg; i.p.) challenge caused loss of VE-cadherin expression within 6 h and expression returned to baseline at 24 h after LPS challenge in WT mice (Fig. 3f). In contrast, LPS challenge had no effect on VE-cadherin expression in $Chfr^{\Delta EC}$ mice (Fig. 3f), indicating EC-expressed $Chfr$

**Fig. 1 | E3 ligase CHFR interaction with VE-cadherin (CDH5) and ubiquitylation of VE-cadherin. a** Schematic of work flow determining the binding of soluble C-terminal green fluorescent protein (GFP)-fused *h*VE-cadherin to human recombinant proteins. **b** Volcano plot showing binding of *h*VE-cadherin-GFP (mean fluorescence intensity vs. *p*-value) to proteins in the microarray. Red dots show positive values and blue dots show negative values. **c** Scattered plot identifying the top 9 ubiquitin E3 ligases binding to *h*VE-cadherin. **b, c** shown are mean values (unpaired Student's *t* test). **d** Schematics of domain structures of human wildtype (WT) CHFR and CHFR mutants lacking forkhead-associated domain (ΔFHA-CHFR), RING finger domain (ΔRF-CHFR), cysteine-rich domain (ΔCR-CHFR), or poly-ADP ribose binding zinc-finger domain (ΔPBZ-CHFR). N-terminal eGFP-tagged WT-CHFR, ΔFHA-CHFR, ΔRF-CHFR, ΔCR-CHFR, and ΔPBZ-CHFR were generated in a pEGFP-C2 expression vector for this study. **e** Human dermal microvascular endothelial cells (HMEC, an endothelial cell line) were transfected with N-terminal GFP tagged WT-CHFR, ΔFHA-CHFR, ΔRF-CHFR, ΔCR-CHFR, or ΔPBZ-CHFR (1.5 μg/ml).

The cells were incubated at 48 h with MG132 (10 μM) and the lysates were used for immunoblot (IB) analysis. **f** Transfected HMEC were used for anti-GFP-agarose beads pull-down assays to study interactions of WT CHFR and CHFR mutants with VE-cadherin. Bottom panel shows quantification of CHFR binding to VE-cadherin as ratio of VE-cadherin (VE-cad)-to-GFP-CHFR. arb. units, arbitrary units. **g** HEK293 cells transfected with HA-tagged ubiquitin (HA-Ub) (0.5 μg/ml) alone or co-transfected with WT-CHFR (1.5 μg/ml), ΔFHA-CHFR (1.5 μg/ml), or ΔRF-CHFR (1.5 μg/ml) were used to study ubiquintylation of VE-cadherin. The cells at 24 h were infected with recombinant adenovirus expressing C-terminal GFP-tagged *m*VE-cadherin (5 pfu/cell). At 24 h thereafter the cells were pretreated with MG132 (10 μM) for 4 h and cell lysates were used for IB analysis. In (**h**), the cell lysates were immunoprecipitated with anti-VE-cadherin antibody and blotted with antibody specific to $K^{48}$-linked or $K^{63}$-linked poly-Ub. Blots were re-probed with antibody specific to either VE-cadherin or CHFR. Representative results from two independent experiments are shown.

mediated degradation of VE-cadherin through ubiquitylation in response to LPS.

To assess ubiquitylation induced by CHFR in mediating the degradation of VE-cadherin, we first determined ubiquitylation pattern of VE-cadherin in *Chfr*^*fl/fl*^ *vs. Chfr*^*ΔEC*^ mice. Ubiquitylation of VE-cadherin via $K^{48}$-linked poly-Ub chains was seen at baseline and post-LPS challenge in WT mice (Fig. 3g). In contrast, both baseline and LPS-induced ubiquitylation of VE-cadherin via $K^{48}$-linked poly-Ub chains were abrogated in the *Chfr*^*ΔEC*^ mice (Fig. 3g). We also observed negligible VE-cadherin ubiquitylation via $K^{63}$-linked poly-Ub chains in *Chfr*^*ΔEC*^ mice (Fig. 3g), suggesting lack of involvement of $K^{63}$-linked poly-Ub in the mechanism of VE-cadherin degradation. Importantly we observed an inverse relationship between VE-cadherin expression and VE-cadherin ubiquitylation via $K^{48}$-linked poly-Ub chains (Fig. 3g), showing the essential role of *Chfr*-mediated ubiquitylation of VE-cadherin via $K^{48}$-linked poly-Ub chains in mediating VE-cadherin degradation.

### Endothelial cell *Chfr* deletion in mice suppresses vascular inflammation

Since CHFR expressed in endothelial cells may be involved in the pathogenesis of inflammation secondary to control of the AJ barrier, we next used *Chfr*^*ΔEC*^ mice to determine the changes in lung endothelial permeability in response to inflammatory stimuli. Histopathological examination of lungs from WT and *Chfr*^*ΔEC*^ mice after LPS challenge (10 mg/kg bw, i.p.) showed that *Chfr*^*ΔEC*^ mice exhibited markedly reduced inflammatory PMN infiltration in lungs (Fig. 4a). We also observed that bronchoalveolar lavage fluid (BALF) from *Chfr*^*ΔEC*^ mice showed reduced protein content and number of infiltrating neutrophils and MPO activity as compared to WT mice (Fig. 4b). In addition, LPS-induced PMN extravasation into lungs (Fig. 4c) and lung vascular permeability (Fig. 4d) were markedly reduced in *Chfr*^*ΔEC*^ mice as compared with WT. Furthermore, the PAR-1(the dominant thrombin receptor expressed in endothelial cells)-, TNFα-, and VEGF-induced increases in vascular permeability were also prevented in *Chfr*^*ΔEC*^ mice as compared with WT (Supplementary Fig. 3a–c). Mortality was reduced in *Chfr*^*ΔEC*^ mice challenged with LPS (Fig. 4e) and cecal ligation puncture a severe model of polymicrobial sepsis (CLP) as compared to WT (Fig. 4f). Next, we determined the expression of PMN adhesive molecules and found no significant difference in the LPS-induced expression of transcripts for ICAM-1, E-selectin, and P-selectin between WT and *Chfr*^*ΔEC*^ mice lung ECs (Supplementary Fig. 4a). We also found no difference in the LPS-induced expression of these adhesive proteins between control and CHFR-depleted ECs (Supplementary Fig. 4b). Since matrix metalloproteinases (MMPs) MMP3 and MMP9 promote lung vascular injury in mice[30–32], we determined expression of MMP3 and MMP9 in response to LPS and found that LPS-induced expression of transcript for MMP3 but not MMP9 was reduced in lung ECs of *Chfr*^*ΔEC*^ mice as compared with WT (Supplementary Fig. 4a). There was no difference seen in the expression of MMP3 and MMP9 between

control and CHFR-depleted ECs in response to LPS (Supplementary Fig. 4b).

### Suppression of *Pseudomonas aeruginosa*-induced pneumonia in *Chfr*^*ΔEC*^ mice

To address further the role of endothelial cell-expressed CHFR in a clinically relevant model of lung injury, we studied pneumonia lung injury[33]. Infection was induced by *Pseudomonas aeruginosa* (PA) airway instillation[34]. We observed strikingly that PA (GFP-PA01; $10^5$ CFU/ mouse, i.t. infection-induced lung vascular hyper-permeability was markedly reduced in *Chfr*^*ΔEC*^ mice as compared to WT (Fig. 5a). Since PMN recruitment is essential for the host response in PA infection[34], we measured the influx of PMN across lung capillaries into lung tissue by in vivo imaging using 2-photon microscopy[35,36]. PA ($10^7$ CFU/mouse) was similarly i.t. instilled into *Chfr*^*ΔEC*^ as well as WT mice, and changes in extravascular and intravascular PMN in lungs were monitored. WT mice showed increased transmigrated PMN into tissue (Fig. 5b); however, in *Chfr*^*ΔEC*^ mice showed significantly reduced PMN transmigration and greater vascular accumulation of PMN (Fig. 5b) consistent with the maintenance of the endothelial junctional barrier in these mice. Under basal conditions, however, we did not see any significant number of PMN in extravascular tissue in either *Chfr*^*ΔEC*^ or WT mice. We observed 40% mortality in WT mice within 3 days of PA (i.t.) dose of $10^6$ PFU/ mouse (Fig. 5c) whereas mortality was not seen in *Chfr*^*ΔEC*^ mice (Fig. 5c). Furthermore, using a lethal model i.t. PA ($10^7$ PFU/ mouse), we observed 100% mortality in WT mice within 24 h (Fig. 5c) whereas only 60% mortality was seen during the same period in *Chfr*^*ΔEC*^ mice (Fig. 5c). Importantly, 20% of *Chfr*^*ΔEC*^ mice survived after PA lethal dose up to 10 days, the longest period monitored (Fig. 5c).

### Endothelial CHFR regulates cell proliferation

Since CHFR regulates the cell cycle by delaying entry into metaphase[37], we studied whether endothelial expressed CHFR influenced EC proliferation. Lung ECs from *Chfr*^*ΔEC*^ mice showed enhanced proliferation as compared with WT mice (Supplementary Fig. 5a, b). siRNA mediated depletion of CHFR in HLMVEC also increased endothelial proliferation (Supplementary Fig. 5c). Thus, CHFR depletion not only induced endothelial barrier integrity as shown above but also enhanced EC proliferation.

### FoxO1 transcriptionally regulates CHFR expression

To address transcriptional mechanisms of CHFR expression, we analyzed the 5'-regulatory regions of mouse and human genes encoding CHFR, and observed multiple binding sites for transcription factors FoxO1 and NF-κB (Fig. 6a). By promoter sequence analysis, we identified that FoxO1 was a NF-κB target gene (Fig. 6a), and thus focused on the central role of FoxO1 in regulating CHFR expression. We observed maximal expression of FoxO1 mRNA and protein at 6 h post-LPS challenge in WT mice (Fig. 6b, c).

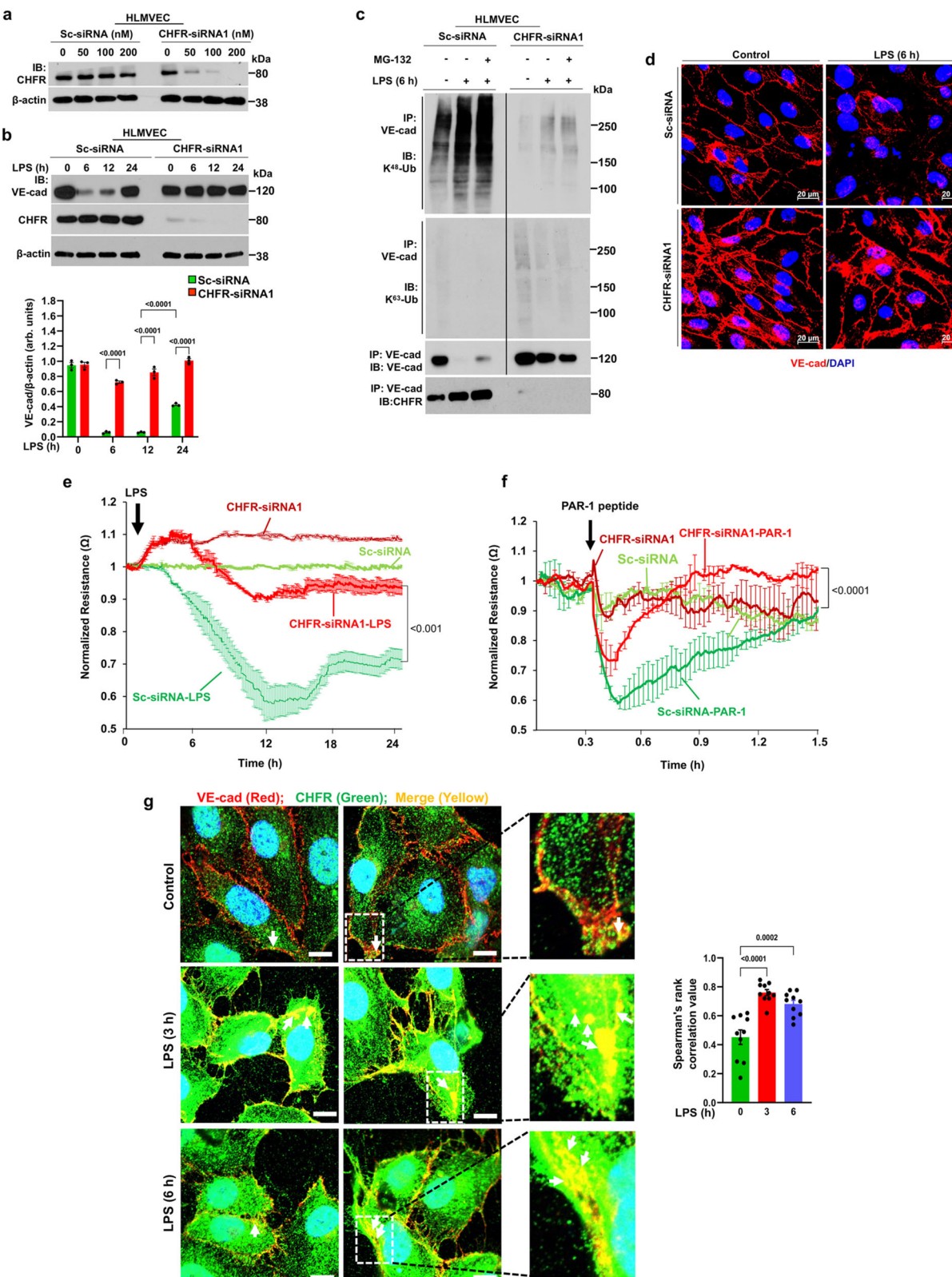

We performed the ChIP assay using lung ECs from WT mice to determine FoxO1 binding to the *mChfr* promoter. Minimal interaction of FoxO1 to the proximal FoxO1 binding sites (−447, −187) was seen in response to LPS whereas several 100-fold increased FoxO1 binding to FoxO1 sites at −578, −566, and −495 of *mChfr* promoter was seen (Fig. 6d). Furthermore, LPS-induced FoxO1 expression amplified CHFR transcription through binding to FoxO1 sites at −578, −566, and −495 of

*mChfr* promoter in ECs. We did not observe any FoxO1 binding to other distal binding sites (−1822, −1643, −944, −741).

As LPS challenge caused time-depended increases in the expression of CHFR mRNA and protein in WT mice (Fig. 6e, f), we determined whether FoxO1 was responsible for the *Chfr* expression. We used the FoxO1 inhibitor AS184285 to study FoxO1 function[38]. In vivo administration of AS184285 significantly reduced LPS-induced expression of

**Fig. 2 | CHFR depletion in endothelial cells prevents VE-cadherin ubiquitylation, degradation, and endothelial barrier breakdown. a** HLMVEC were transfected scrambled-siRNA (Sc-siRNA) or CHFR-siRNA1. At 72 h post-transfection, cells were used for IB analysis (*n* = 2 independent experiments). **b** HLMVEC transfected with 100 nM of either Sc-siRNA or CHFR-siRNA1 were challenged with LPS (5 μg/ml) for the indicated times and cell lysates were used for IB. Shown are mean values ± SEM (*n* = 3 independent experiments; two-way ANOVA followed by Tukey's post-hoc test). **c** HLMVEC transfected with Sc-siRNA or CHFR-siRNA1 as above were preincubated with or without proteasomal inhibitor MG132 (10 μM) for 2 h, and exposed to LPS. Cell lysates were immunoprecipitated (IP-ed) with anti-VE-cad pAb and blotted with antibodies specific to K[48]-linked poly-Ub (K[48]-Ub) or K[63]-linked poly-Ub (K[63]-Ub) (*n* = 2 independent experiments). **d** cells were stained with anti-VE-

cad pAb. red, VE-cadherin (VE-cad); blue, DAPI. **e, f** cells were used to measure LPS- or PAR-1 activating peptide (TFLLRNPNDK-NH$_2$; 25 μM)-induced changes in TER as a measure of endothelial permeability changes. Arrow indicates time of LPS, PAR-1 peptide, or media addition. Shown are mean values ± SEM (*n* = 3 independent experiments; unpaired two-tailed Student's *t* test). **g** Confluent HLMVEC exposed to LPS for indicated time intervals were stained with antibodies to VE-cadherin (red), CHFR (green), and DAPI (blue). Confocal images were used to assess co-localization of VE-cadherin with CHFR by Spearman's rank correlation. Shown are mean values ± SEM (*n* = 10 field of view/group; one-way ANOVA followed by Tukey's post-hoc test). *Right panel* shows magnified images. Arrows show co-localization of CHFR with VE-cadherin at AJs or in endocytosed vesicles. Red, VE-cadherin; Green, CHFR; Blue, DAPI; Scale bar = 10 μm.

CHFR (Fig. 6g) whereas NF-κB-dependent ICAM-1 expression was unaffected (Fig. 6g). To address the function of FoxO1-mediated CHFR expression in the pathogenesis of ARDS in patients, we examined protein expression of CHFR and FoxO1 in lung sections from controls and non-surviving ARDS patients. De-identified lung tissue obtained at autopsy was paraffin embedded and used for immunohistochemistry. We observed augmented expression of both CHFR (Fig. 6h) and FoxO1 (Fig. 6i) in lung ECs of non-surviving ARDS patients as compared with non-ARDS controls (Fig. 6h, i).

### FoxO1 deletion in endothelial cells prevents expression of CHFR and mitigates vascular inflammation

To determine the role of EC-expressed FoxO1 in regulating CHFR expression and thereby vascular barrier integrity, we deleted *FoxO1* in adult mouse EC using the *Cre*-dependent Cas9 knock-in mouse model (Fig. 7a)[39]. We first generated CRISPR/Cas9-cdh5-Cre mice and *FoxO1* was deleted in ECs by liposome-mediated in vivo delivery of the plasmid (pGS) encoding a single guide-RNA (sgRNA) targeting *mFoxO1*[35]. We used two different sgRNAs (*FoxO1*-sgRNA-1; *FoxO1*-sgRNA-2) to target *mFoxO1* or a scrambled sgRNA (Sc-sgRNA) as control (Fig. 7b). Administration of plasmid encoding *mFoxO1*-sgRNA-1 disrupted *FoxO1* expression in lung ECs (Fig. 7b, c). LPS challenge increased the expression of FoxO1 (Fig. 7d) as well as CHFR (Fig. 7e) in WT (Sc-sgRNA-treated) mice (Fig. 7d, e) whereas LPS failed to alter expression of either protein in FoxO1-sgRNA-1 (*FoxO1*$^{ΔEC}$) mice (Fig. 7d, e) and induce the degradation of VE-cadherin (Fig. 7f).

In a control experiment, we observed that *FoxO1*$^{ΔEC}$ mice showed inhibition of expression of the FoxO1 target gene angiopoietin 2 (Ang-2) (Fig. 7f). In line with this result, LPS failed to ubiquitylate VE-cadherin via K[48]-linked poly Ub chains in *FoxO1*$^{ΔEC}$ mice (Fig. 7g). We also assessed LPS-induced endothelial barrier breakdown in *FoxO1*$^{ΔEC}$ mice. In *FoxO1*$^{ΔEC}$ mice, the LPS-induced increase in lung vascular permeability was abrogated (Fig. 7h). Consistent with this observation, CLP-induced mortality was also markedly reduced in *FoxO1*$^{ΔEC}$ mice as compared to WT (Fig. 7i). These results together showed that FoxO1-mediated CHFR expression in EC induced endothelial barrier breakdown through the degradation of VE-cadherin and opening the endothelial junctional barrier (model in Fig. 7j).

## Discussion

VE-cadherin expressed at endothelial AJs functions as a "gatekeeper" to restrict free extravasation of plasma proteins and migration of blood cells into tissue. Loss of VE-cadherin expression at AJs in contrast is a primary factor in the pathogenesis of vascular injury and leaky vessels in inflammatory disease. The ubiquitin-proteasome system (UPS) is crucial mechanism for the quality control of cellular proteins[23–25] but little is known about UPS-mediated VE-cadherin degradation in endothelial cells during the normal turnover cycle of VE-cadherin and in response to inflammation associated with disruption of the VE-cadherin junctional barrier. To define the role of UPS in regulating endothelial barrier integrity, on performing an unbiased protein interaction analysis we identified the E3-ligase CHFR as the primary VE-

cadherin degrading protein. EC-expressed CHFR was shown to be required for ubiquitylation and subsequent degradation of VE-cadherin. Gene knockdown studies and EC-restricted gene knockout mouse models furthermore established the function of CHFR in regulating endothelial barrier integrity through the controlling VE-cadherin expression at AJs. Loss of CHFR in endothelial cells prevented the breakdown of VE-cadherin junctions and preserved endothelial integrity induced by agents such as LPS known to degrade VE-cadherin[6,7,16,21]. Together our findings uncovered the crucial role of CHFR in ubiquitylation of VE-cadherin and in thereby regulating endothelial barrier integrity at the level of AJs.

A previous study showed VE-cadherin degradation occurred via proteasomes in endothelial cells[40] consistent with present observations. In addition, other E3 ligases may also be involved. Kaposi sarcoma virus-encoded E3-ligase K5 (MARCH-family E3-ligase) induced ubiquitylation and degradation of VE-cadherin in endothelial cells[41], which was coupled to vascular leakage and tumorigenesis[42]. Another study showed that bradykinin-induced phosphorylation of VE-cadherin at Y658 and Y685 caused VE-cadherin internalization, ubiquitylation, and co-localization with K[63]-linked ubiquitin in the internalized vesicles in ECs[14]. In the present study, we describe the central role of the E3-ligase CHFR in regulating the integrity of endothelial barrier through controlling the degradation of VE-cadherin at AJs. We also showed the involvement of CHFR in mediating LPS-induced disruption of VE-cadherin junctional barrier. We observed that LPS-induced CHFR dependent K[48]-linked polyubiquitylation of VE-cadherin, thereby triggering degradation of VE-cadherin at AJs. Endothelial E3-ligase CHFR was thus crucial in maintaining VE-cadherin junctional homeostasis in the basal state and in response to inflammatory stimuli such as LPS through controlling ubiquitylation state of VE-cadherin.

To show the relevance to human vessels we studied the role of endogenously expressed CHFR in regulating VE-cadherin function in human lung microvascular endothelial cells. Knockdown of CHFR prevented basal as well as LPS-induced K[48]-linked VE-cadherin ubiquitylation and degradation and endothelial barrier breakdown in response to LPS. It was shown that K[63]-linked ubiquitylation of VE-cadherin in response to bradykinin[14]; however, we did not observe K[63]-linked ubiquitylation of VE-cadherin in either the basal state or post-LPS challenge, suggesting that inflammatory mechanisms can induce endothelial junctional breakdown through activating distinct patterns of VE-cadherin ubiquitylation.

To address the in vivo role of EC-expressed CHFR, we generated EC-restricted *Chfr* knockout (*Chfr*$^{ΔEC}$) mice. The defining phenotype of these mice was augmented expression of VE-cadherin and recruitment of its binding partners VE-PTP, p120-catenin, and β-catenin in endothelial cells. The induction of endotoxemia failed to induce K[48]-linked ubiquitylation and degradation of VE-cadherin in endothelial cells of *Chfr*$^{ΔEC}$ mice. In agreement with this finding, we also showed that LPS-induced vascular permeability and PMN influx into lung tissue were markedly reduced in *Chfr*$^{ΔEC}$ mice as compared with wild-type. Moreover, using other means of increasing junctional permeability, the

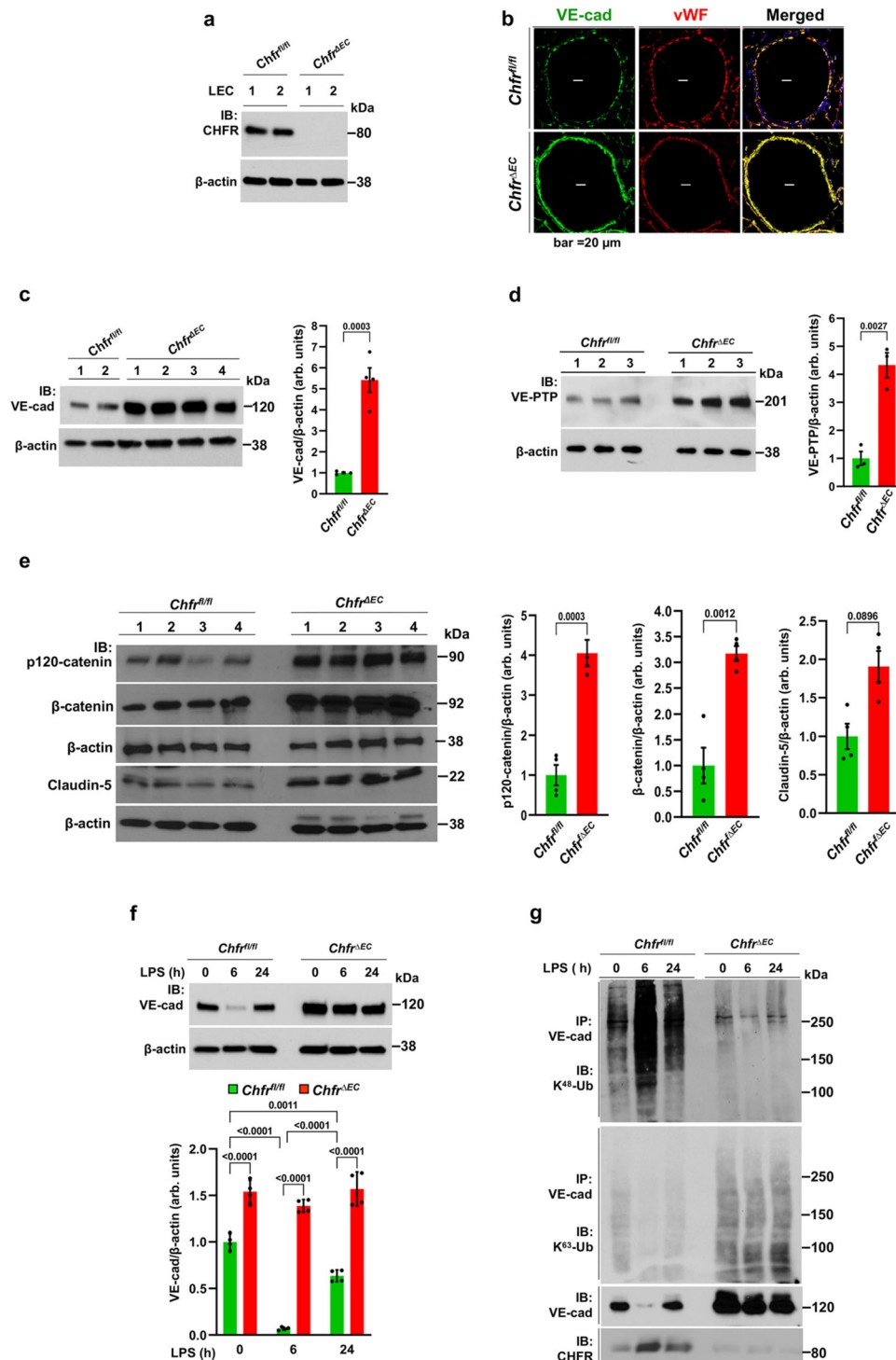

**Fig. 3 | Endothelial cell-restricted *Chfr* deletion in mice prevents VE-cadherin ubiquitylation and augments expression of VE-cadherin. a** Lung ECs (LEC) from *Chfr^ΔEC^* mice showed no *Chfr* expression as compared with LEC from *Chfr^fl/fl^* mice. **b** lung sections stained with antibodies specific to VE-cadherin (VE-cad, green) and vWF (EC-marker, red). **c**–**e** IB analysis of lung tissues showed augmented expression of VE-cad (**c**; *n* = 4 mice/genotype), VE-PTP (**d**; *n* = 3 mice/genotype), p120-catenin, β-catenin, and claudin-5 (**e**; *n* = 4 mice/genotype) in *Chfr^ΔEC^* mice. Shown are mean values ± SEM (unpaired two-tailed Student's *t* test). **f** *Chfr^fl/fl^* and *Chfr^ΔEC^* mice

injected i.p. with LPS (10 mg/kg body weight) for 0, 6, and 24 h were used to measure VE-cad expression. Shown are mean values ± SEM (*n* = 4 mice/genotype/ group; two-way ANOVA followed by Tukey's post-hoc test). **g** Lung tissue harvested from *Chfr^fl/fl^* and *Chfr^ΔEC^* mice after LPS i.p. as above was used to assess VE-cad ubiquitylation. Lung tissue lysates were immunoprecipitated (IP-ed) with anti-VE-cad pAb and immunoblotted with mAb specific to K^48^-linked poly-Ub (K^48^-Ub) or K^63^-linked poly-Ub chain (K^63^-Ub). Blots were re-probed with antibody specific to VE-cad or CHFR. Shown is a representative blot (*n* = 2 independent experiments).

thrombin receptor PAR-1-activating peptide, TNF-α-, and VEGF, we showed all increased junctional permeability dependent on endothelial expressed CHFR. We also observed that LPS- or CLP-induced mortality was reduced in *Chfr^ΔEC^* mice consistent with the maintenance

of normal endothelial barrier function in providing survival advantage in mice in inflammatory diseases that would otherwise be fatal.

We also addressed the role of CHFR in regulating the integrity of lung endothelial junctions in response to live bacterial

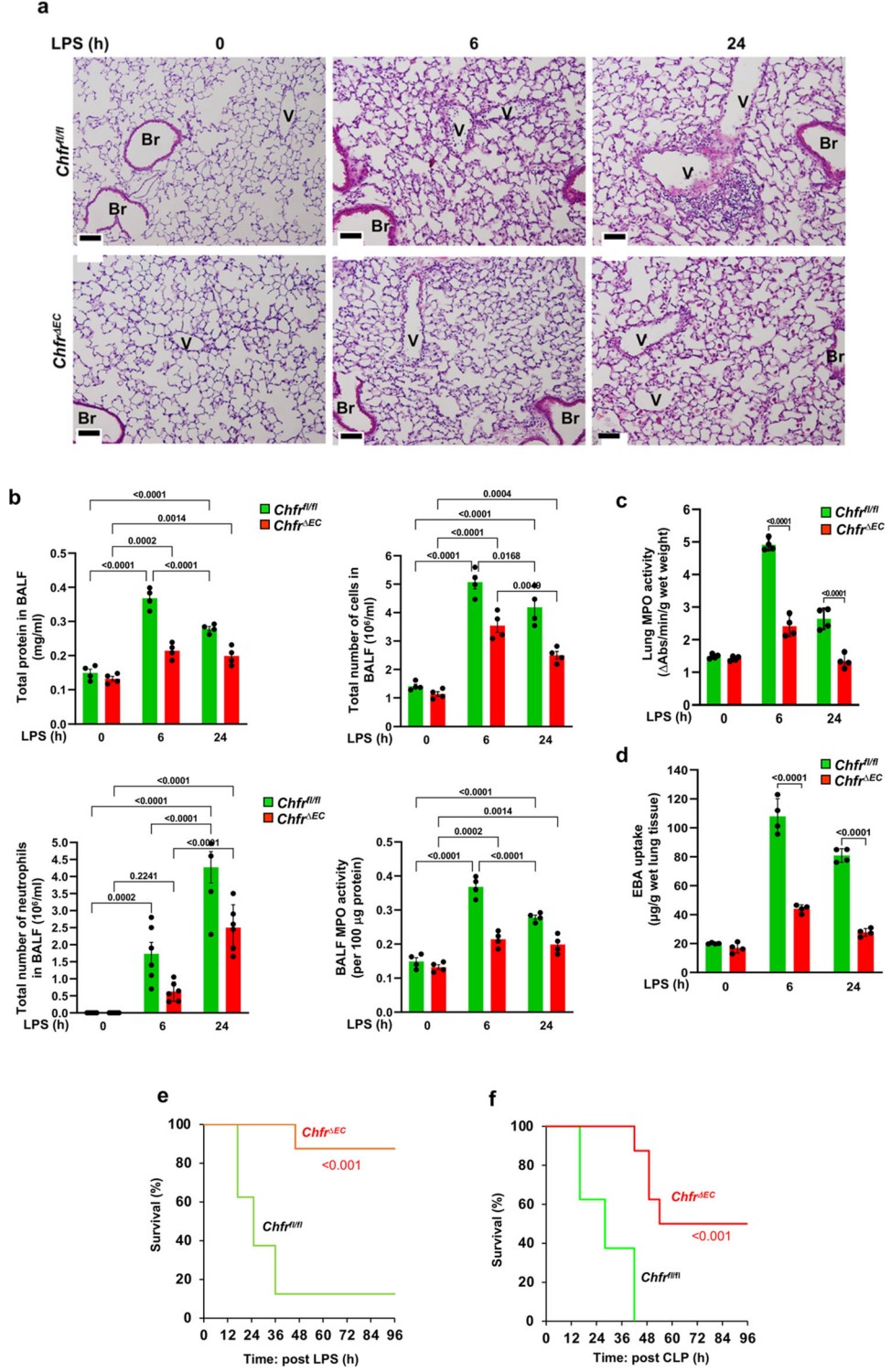

**Fig. 4 | *Chfr* deletion in endothelial cells of mice suppresses LPS-induced vascular injury and mortality. a** Hematoxylin and eosin staining of lung sections from *Chfr^fl/fl^* and *Chfr^ΔEC^* mice injected i.p. with LPS (10 mg/kg body weight) for 0, 6, and 24 h. Scale bar, 50 µm. Br, bronchus; V, vessel. **b** Protein contents, total cells, PMNs, and MPO activity in bronchoalveolar lavage fluid (BALF) from *Chfr^fl/fl^* and *Chfr^ΔEC^* mice as in (**a**). **c** Lungs harvested at indicated time points after LPS i.p. (10 mg/kg body weight) were used for myeloperoxidase activity to assess PMN influx. **d** *Chfr^fl/fl^*

and *Chfr^ΔEC^* mice injected i.p. with LPS (10 mg/kg body weight) for 0, 6, and 24 h were used to assess lung vascular leak by measuring Evans blue bound albumin (EBA) uptake in lungs. **b–d** results shown are mean values ± SEM (*n* = 4 mice/genotype/group; two-way ANOVA followed by Tukey's post-hoc test). **e** Survival of age- and weight-matched *Chfr^fl/fl^* and *Chfr^ΔEC^* mice injected i.p. with LPS (10 mg/kg body weight). **f** Survival of age- and weight-matched *Chfr^fl/fl^* and *Chfr^ΔEC^* mice challenged with CLP (**e**, **f** *n* = 8 mice/genotype; *Chfr^fl/fl^* vs *Chfr^ΔEC^*; log-rank test).

challenge. We observed that *Pseudomonas aeruginosa* (PA) infection-induced lung vascular permeability and PMN transendothelial migration in lung capillaries were markedly reduced in *Chfr^ΔEC^* mice as compared with WT mice. As in the above studies in

the endotoxemia model, these results showed the central role of CHFR in regulating endothelial barrier integrity through controlling the expression of VE-cadherin in endothelial cells in PA infection.

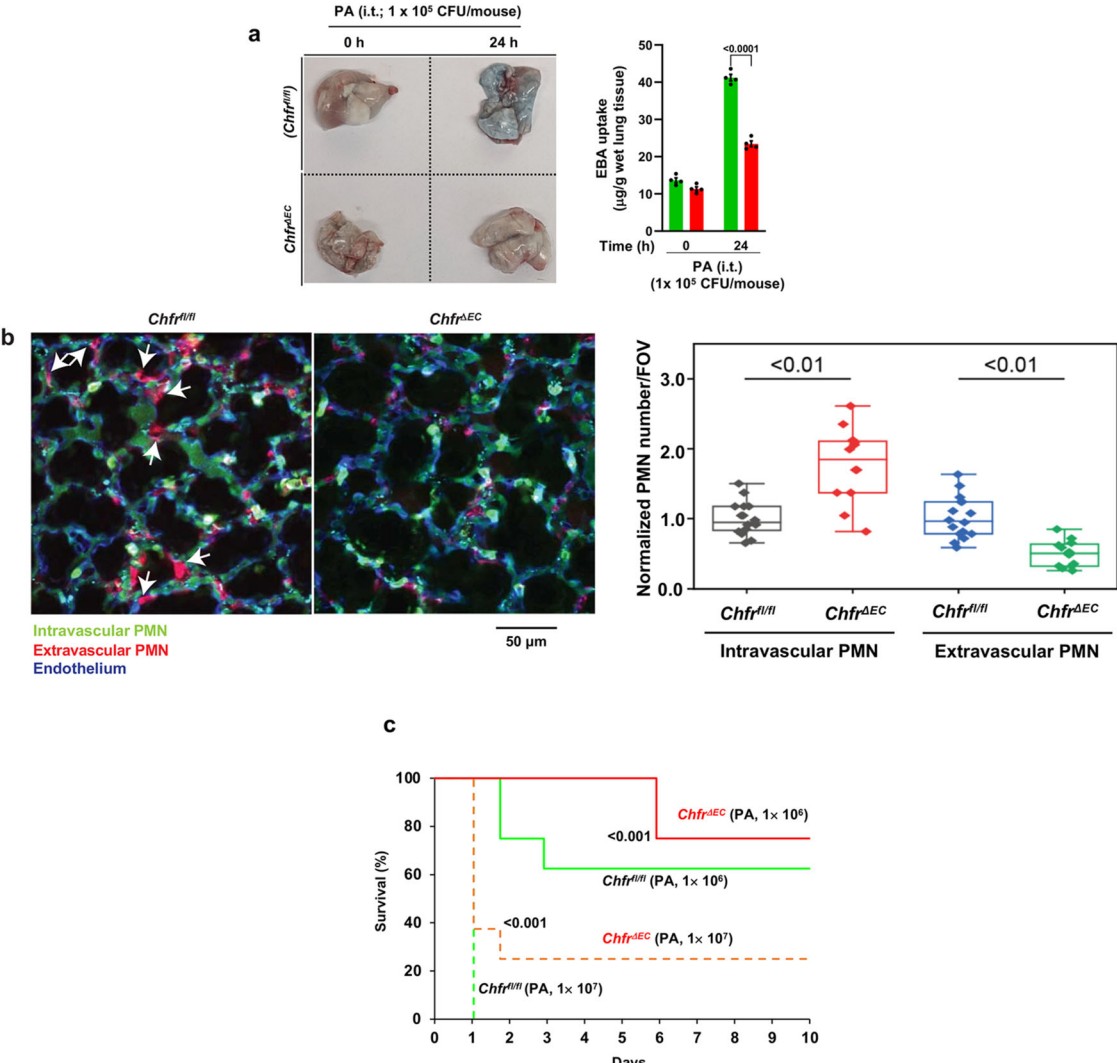

**Fig. 5 | *Chfr* deficiency in endothelial cells of mice mitigates *Pseudomonas aeruginosa* infection-induced lung vascular barrier breakdown, PMN transmigration, and mortality. a** *Chfr^{fl/fl}* and *Chfr^{ΔEC}* mice i.t. *Pseudomonas aeruginosa* (PA) instilled (10^5 CFU/mouse) for 0 and 24 h were used to assess lung vascular leak by measuring EBA uptake in lungs. Representative lung images are shown on *left* and quantified results are shown on *right*. Shown are mean values ± SEM (n = 4 mice/genotype/group; two-way ANOVA followed by Bonferroni's post-hoc test). **b** *Chfr^{fl/fl}* and *Chfr^{ΔEC}* mice i.t. instilled lethal dose of PA (10^7 CFU/mouse) were used for imaging with 2-photon microscope 6 h after i.t. PA challenge. Alexa 594-labeled Ly6G Ab (20 µg/mouse) i.v. injected prior to i.t. *PA*. At 5 min before mice were surgically prepared for imaging, Brilliant Violet 421-labeled LY6G Ab (10 µg/mouse)

and EC marker Ab (SeTau647-labeled CD31 Ab, 20 µg/mouse) were i.v. injected. Arrows indicate transmigrated PMNs. Scale bar, 50 µm. Quantitative analysis of intravascular and extravascular PMN numbers shown in *right panel*. Intravascular PMN per field of view (FOV) in WT were normalized as 1. Shown are mean values ± SEM (*n* = 5 mice/genotype; two-way ANOVA followed by Tukey's post-hoc test). In box and whisker plots, the box is determined by 25–75 percentile and the whisker is determined by 5 to 95 percentile. The cenral line represents the median value. **c** Survival of age- and weight-matched *Chfr^{fl/fl}* and *Chfr^{ΔEC}* mice challenged with two different doses of PA (i.t. instilled; 10^6 pfu/mouse or 10^7 pfu/mouse). (*n* = 8 mice/genotype; *Chfr^{fl/fl}* vs *Chfr^{ΔEC}*; log-rank test).

The present studies help to inform how CHFR expression is transcriptionally regulated during vascular inflammation in endothelial cells. Performing transcription factor binding motif analysis, we observed that CHFR was especially rich in FoxO1 binding sites in the promoter regions of both mouse and human CHFR genes. FoxO1 was shown to bind the *Chfr* promoter following LPS challenge of endothelial cells. Moreover, FoxO1 inhibition using AS184285 in vivo suppressed CHFR expression consistent with the role of FoxO1 as crucial for upregulation of CHFR expression. To address the in vivo role of FoxO1, we deleted *FoxO1* in adult mouse endothelial cells in which EC-restricted FoxO1 knockout (*FoxO1^{ΔEC}*) showed defective LPS-induced expression of FoxO1 as well as CHFR and LPS failed to ubiquitylate and degrade VE-cadherin. Furthermore, endotoxemia-induced lung vascular leak and mortality were markedly reduced as compared with wild-type. These results together showed that FoxO1-mediated CHFR

expression in endothelial cells mediates K^48-linkage specific polyubiquitylation of VE-cadherin triggering endothelial barrier breakdown at AJs and setting the stage for induction of inflammation.

Besides our observation that FoxO1 transcriptionally upregulated CHFR, FoxO1 is also known to upregulate the expression of several genes such as Ang-2[43–47] as well silence claudin-5 expression through interacting with Tcf-4/β-catenin transcriptional repressor complex in endothelial cells[46]. Ang-2 functions as a Tie-2 receptor antagonist and its amplified expression during vascular inflammation promoted endothelial barrier breakdown[47]. Furthermore, LPS-induced FoxO1 expression may itself contribute to lung vascular injury through the induction of autophagy and directly downregulating VE-cadherin expression in endothelial cells[48]. Another study consistent with the present observation showed that FoxO1-mediated MMP3 expression contributed to LPS-induced lung vascular injury in mice[31]. We also

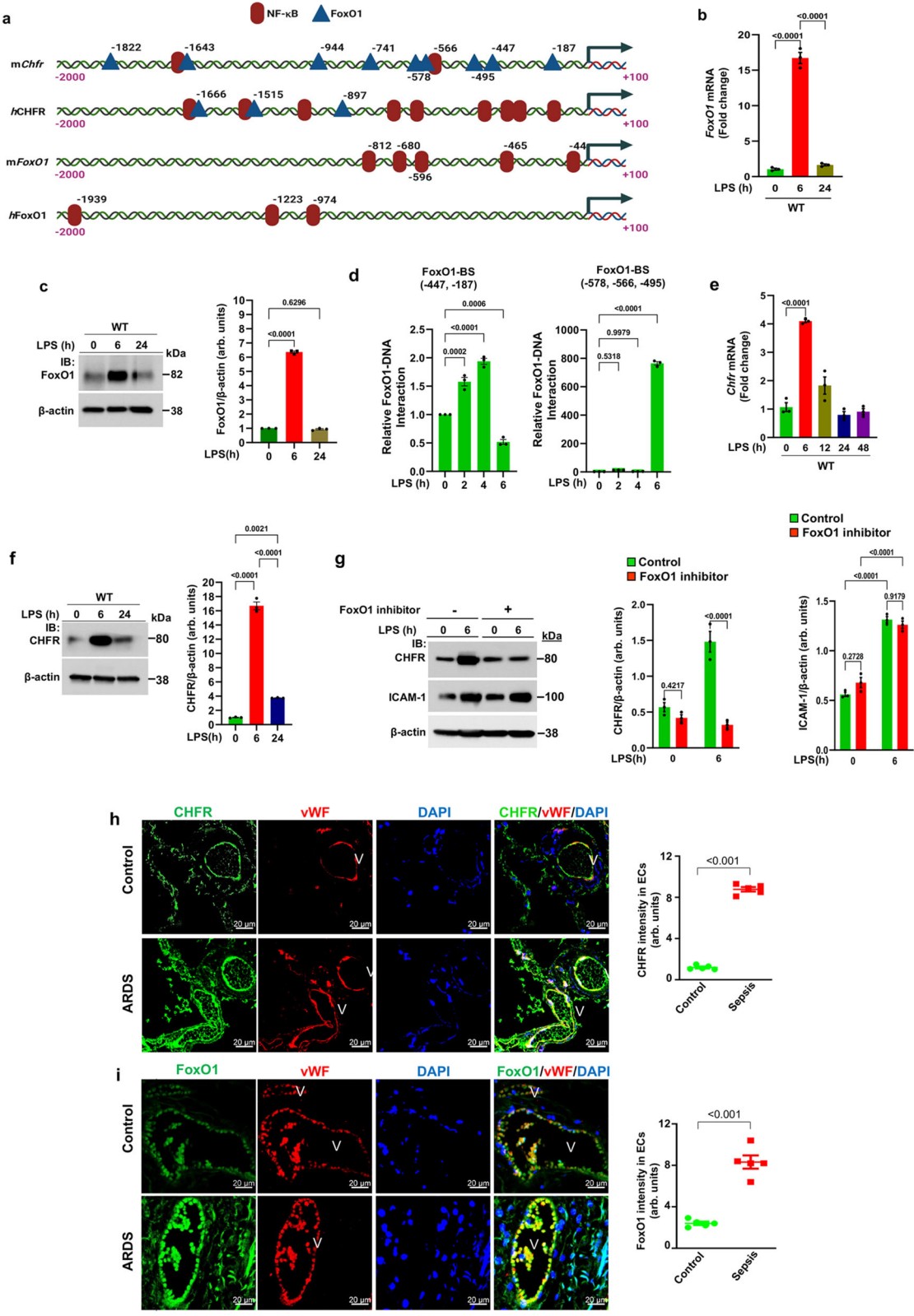

observed that LPS-induced MMP3 expression was abrogated in $Chfr^{\Delta EC}$ mouse endothelial cells, which may be due to VE-cadherin/PI(3)K/AKT axis-mediated inhibition of FoxO1 function[46].

VE-cadherin expression at endothelial AJs not only contributes to endothelial barrier integrity, but also modulates the expression of genes involved in endothelial stability[46,49]. Recently, it was shown that VE-cadherin expression and clustering at cell-cell junctions

upregulated endothelial stability-related genes such as VE-PTP, von Willebrand factor (vWF), and claudin-5 through inhibiting polycomb repressive complex-2 (PRC2) nuclear activity by sequestering PRC2's catalytic subunit enhancer of zeste homolog 2 (Ezh2) and preventing the FoxO1/β-catenin complex nuclear translocation[49]. Thus, these reported findings support the emerging concept that E3 ligase CHFR-mediated degradation of VE-cadherin may promote the PRC2

**Fig. 6 | FoxO1 signaling downstream of TLR4 induces CHFR expression.**
**a** Schematic showing putative transcription factor binding sites in the promoter regions of *mChfr*, *h*CHFR, *m*FoxO1, and *h*FoxO1. **b**, **c** WT mice were challenged with LPS (i.p; 10 mg/kg body weight) for different times. **b** Lungs harvested were used for total RNA isolation followed by RT-qPCR. Shown are mean values ± SEM (*n* = 3 mice/group, one-way ANOVA followed by Tukey's post-hoc test). **c** Lungs were used for IB to determine expression of FoxO1. Shown are mean values ± SEM (*n* = 3 mice/group; one-way ANOVA followed by Tukey's post-hoc test). **d** WT mice lung ECs challenged with LPS (5 μg/ml) for different time intervals were used for ChIP assay to determine FoxO1 binding to *mChfr* promoter. Shown are mean values ± SEM. FoxO1-BS, FoxO1 binding sites; (*n* = 3 independent experiments; one-way ANOVA followed by Tukey's post-hoc test). **e**, **f** WT mice challenged with LPS as in (**b**) and then lungs harvested were used for total RNA isolation followed by RT-qPCR and IB

to determine CHFR expression. Shown are mean values ± SEM (**e**–**f**, *n* = 3 mice/group; one-way ANOVA followed by Tukey's post-hoc test). **g** WT mice were injected with DMSO or the FoxO1 inhibitor (AS1842856; 5 mg/kg, i.p. one injection per day for 4 days) and then challenged with LPS (i.p.;10 mg/kg body weight) or saline for 6 h. Lungs harvested were used for IB analysis to determine expression of *Chfr* and ICAM-1. Results show a representative blot. Shown are mean values ± SEM (*n* = 3 mice/group; two-way ANOVA followed by Tukey's post-hoc test). **h**, **i** Augmented expression of CHFR and FoxO1 in lung endothelial cells of patients with acute respiratory distress syndrome (ARDS). Lung sections from non-ARDS control and sepsis/ARDS patients were stained with EC marker vWF (red) and the E3 ligase CHFR (green) (**h**) or FoxO1 (green) (**i**). DAPI (blue). Shown are mean values ± SEM (*n* = 5 samples from each group; unpaired two-tailed Student's *t* test). Fluorescent intensity is presented as arbitrary units (arb. units). V, blood vessel.

mediated silencing of the vascular stability genes in endothelial cells, which could potentially contribute to vascular inflammatory responses.

In summary, we demonstrated that endothelial cell-deletion of CHFR in mice augmented the integrity of VE-cadherin junctions and prevented breakdown of AJs in response to inflammatory mediators such as LPS, TNF-α, and thrombin. We also observed several-fold increased expression of both CHFR and FoxO1 in lung endothelial cells of non-surviving ARDS patients compared with non-ARDS controls. The latter findings together with studies in mice infected with *Pseudomonas* suggest that unchecked expression of FoxO1 and CHFR in endothelial cells may be essential for the pathogenesis of inflammatory diseases. Thus, endothelial cell-expressed ubiquitin E3-ligase CHFR mediated degradation of VE-cadherin is essential in the mechanism of endothelial barrier breakdown and thereby in the genesis of vascular inflammation.

## Methods
### Primary Antibodies
Rabbit polyclonal antibody (pAb) against the C-terminus of VE-cadherin (catalog #Ab33168; immunoblot [IB], 1:1500; immunostaining [IS], 1:150; immunoprecipitation [IP], 1 μg/100 μg cell lysate protein) was from Abcam. Mouse monoclonal antibody (mAb) against the N-terminus of VE-cadherin (catalog #348502; IB, 1:1000) from BioLegend. Rabbit mAb against FoxO1 (catalog #2880; IB, 1:1000; IS, 1:100), rabbit (mAb) against CHFR (catalog #D40H6; IB, 1:1000; IS, 1:100), and rabbit mAb against ubiquitin K[63]-linkage-specific (catalog #5621S; IB, 1:1000) were from Cell Signaling Technology. Mouse mAb against the C-terminus of VE-cadherin (catalog #sc-9989; IB, 1:1000), goat pAb against vWF (catalog #sc-8068; IS, 1:150), rabbit pAb against VE-PTP (catalog #sc-28905; IB, 1:1000), goat pAb against ICAM-1 (catalog #sc-1511; IB, 1:1000), and rabbit pAb against β-catenin (catalog #sc-7199; IB, 1:1000) were from Santa Cruz Biotechnology. Rabbit mAb against ubiquitin K[48]-linkage-specific (catalog #05-1307; IB, 1:1000) was from Millipore Sigma. Mouse mAb against β-actin (catalog #A5441; IB, 1:2000) was from Sigma-Aldrich Inc. Rabbit pAb against GFP (catalog #50430-2-AP; IB, 1:2000), rabbit pAb against CHFR (catalog #12169-1-AP; IB, 1:1000), rabbit pAb against ICAM-1 (catalog #10831-1-AP; IB, 1:1000), rabbit pAb against E-selectin/CD62E (catalog #20894-1-AP; IB, 1:5000), rabbit pAb against P-selectin/CD62P (catalog #13304-1-AP; IB, 1:500), rabbit pAb against Angiopoietin-2 (catalog #24613-1-AP; IB, 1:1000), rabbit pAb against MMP3 (catalog #17873-1-AP; IB, 1:1000), rabbit pAb against MMP9 (catalog #10375-2-AP; IB, 1:1000), and rabbit pAb against p120-catenin (catalog #12180-1-AP; IB, 1:1000) were from Proteintech. Rabbit pAb against claudin-5 (catalog #34-1600; IB, 1:1000) was from Invitrogen.

### Secondary antibodies
Anti-rabbit IgG (H + L) peroxidase labeled (catalog #5220-0458; IB, 1:2000) and anti-mouse IgG (H + L) peroxidase labeled (catalog #5450-0011; IB, 1:2000) were from Sera Care KPL. Donkey anti-goat IgG (H + L) HRP labeled (catalog #A16005; IB, 1:2000), chicken anti-rabbit IgG

(H + L) Alexa Fluor 594 labeled (catalog # A21442; IS, 1:500), chicken anti-mouse IgG (H + L) Alexa Fluor 647 labeled (catalog #A21463; IS, 1:500), goat anti-rabbit (H + L) Alexa Fluor 488 labeled (catalog #A11034; IS, 1:500), donkey anti-goat IgG (H + L) Alexa Fluor 546 labeled (catalog #A11056; IS, 1:500), and chicken anti-rabbit IgG (H + L) Alexa Fluor 488 labeled (catalog #A21441; IS, 1:500) were from Invitrogen.

### Other reagents
Scrambled-siRNA (Sc-siRNA) and human (*h*)-specific siRNA1 against CHFR (A pool of 3 target-specific siRNAs, sense: 5′-CUCUGUGGCAA-GUGAUGAAtt-3′; sense: 5′-GAAGAAGAUGUGCAAAGUAtt; sense: 5′-GGUAUAGUAUGGUAUUUGAtt-3′) were obtained from Santa Cruz Biotechnology. *h*-specific siRNA2 against CHFR (A pool of 4 target-specific siRNAs, sense 5′-GAACAGUGAUUAACAAGCU-3′; sense 5′-CAACGUGGCAUACCUCUAU-3′; sense 5′-GCACUCAGGUGAAAGCUCA-3′; sense 5′-CAACAACAGCUACGAGUCA-3′) was from Dharmacon. Recombinant adenovirus expressing *h*CDH5-GFP (carboxyl-terminal tagged with GFP) was prepared as described[50]. Recombinant adenovirus expressing *mCdh5-GFP* (carboxyl-terminal tagged with GFP) was from Vector Biolabs (Malvern, PA). PAR-1-activating peptide (TFLLRNPNDK-NH₂) as the C-terminal amide was synthesized with a purity >95% by Genscript (Piscataway, NJ). Peptide purity and amino acid sequences were determined by HPLC and MS, respectively. pGS-gRNA plasmid encoding sgRNAs to target *mFoxO1*gene, control sgRNA, pEGFP-C2 plasmid (Addgene catalog #61853) encoding wild-type human CHFR (WT *h*-CHFR), *h*CHFR mutant lacking forkhead-associated domain (ΔFHA-CHFR), *h*CHFR mutant lacking RING finger domain (ΔRF-CHFR), mutant lacking cysteine-rich domain (ΔCR-CHFR), and mutant lacking poly-ADP ribose binding zinc-finger domain (ΔPBZ-CHFR) were custom prepared by Genscript (Piscataway, NJ). These CHFR deletion constructs were prepared using standard molecular biology methods. DNA sequencing was performed to verify the deletion. PCR primers were custom synthesized by Integrated DNA Technologies (Coralville, IA). HA-tagged ubiquitin-expressing plasmid (catalog #17608) was from Addgene. Vascular Endothelial Growth Factor (VEGF) (catalog #V7259) was from Sigma. TNF-α (2 × 10⁷ U/mg protein), human recombinant (catalog #ALX-520-002) was from Enzo Life Sciences.

### Recombinant *h*CDH5
HEK293 cells (ATCC, catalog #CRL-1573) grown on 60 mm culture dishes were infected with recombinant adenovirus expressing *h*CDH5-GFP (Adeno-VE-Cad-GFP) (5 pfu/cell). 24 h after infection, cells were washed twice with ice-cold PBS and then lysed with lysis buffer (50 mM Tris-HCl, pH7.5, 150 mM NaCl, 1 mM EGTA, 1% Triton X-100, 0.25% sodium deoxycholate, 0.1% SDS, 10 μM orthovanadate, and protease-inhibitor mixture) for 30 min at 4⁰C. Cell lysates were centrifuged at 30,000 × *g*, 4 °C for 10 min. The supernatant containing 2 mg of protein was diluted 4 times with dilution buffer (10 mM Tris-HCl, pH7.5, with 150 mM NaCl, 0.5 mM EGTA) and then the *h*VE-cad-GFP protein

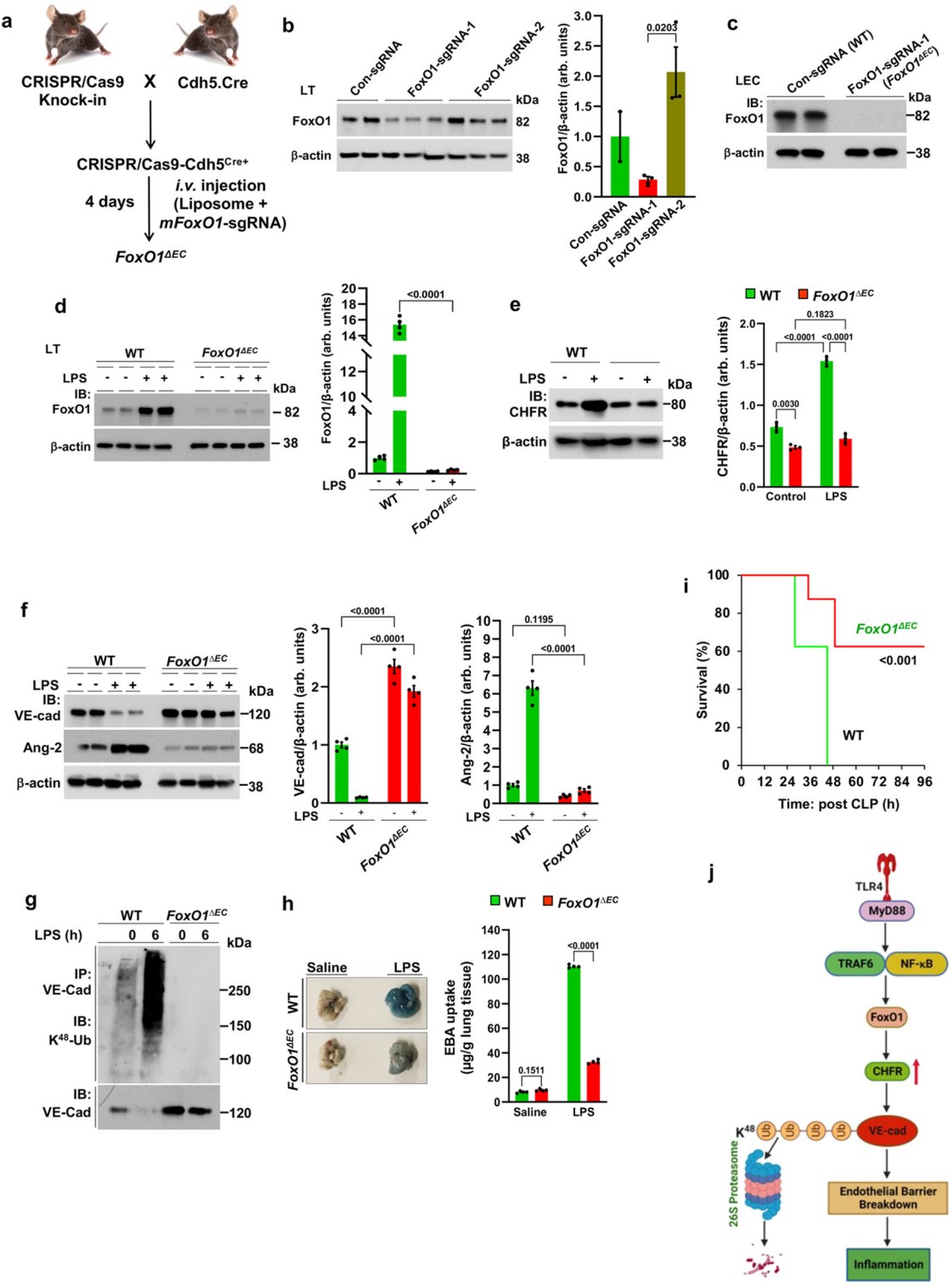

was affinity purified using GFP-Trap Magnetic Agarose beads (ChromoTek; catalog #gtma-10) following the manufacturer's protocol. GFP-Trap Magnetic Agarose beads bound proteins were eluted with elution buffer (200 mM glycine pH 2.5, containing 0.1% Triton X-100). After elution, the sample was neutralized with 1 M Tris pH 10.4, lyophilized, and suspended in PBS containing 0.05% Tween-20 (PBST).

### Identification of CHFR binding to CDH5 (VE-cadherin)

AVM BIOMED Snapshot Proteomics system was used to identify VE-cadherin binding proteins. AVM BIOMED microarray system

containing 21, 065 human recombinant proteins onto thin-film nitrocellulose-coated slides were prepared as described previously[51] by GenTel's Biosciences (Madison, WI). The binding of $h$VE-cad-GFP protein with the microarray was measured by AVM BIOMED. Arrays were blocked for 1 h at room temperature (RT) in PBS containing 0.05% Tween-20 (PBST), 20 mM reduced glutathione, 1 mM DTT, 1% BSA, and 25% glycerol. Identical arrays in parallel were incubated for 90 min at RT with $h$VE-cad-GFP protein (5 μg protein/ml, suspended in PBST). Arrays were washed in 2 changes PBST, PBS, then two changes water before centrifugal

**Fig. 7 | Endothelial-specific FoxO1 deletion mitigates endotoxemia-induced vascular inflammatory response. a** Depicts the protocol used to create EC-restricted *FoxO1* knockout (*FoxO1ΔEC*) mice. **b** CRISPR/Cas9-cdh5-Cre+ mice were injected with plasmid (pGS-gRNA) encoding sgRNAs (sgRNA-1: 5′-CACGGGGGT-CAAGCGGTTCA-3′ or sgRNA-2: 5′- AATTCGGTCATGCCAGCGTA-3′) to target the *mFoxO1* gene or control-sgRNA (con-sgRNA: 5′-GCGAGGTATTCGGCTCCGCG-3′). Lungs were used 4 days after sgRNA treatment for IB analysis. Shown are mean values ± SEM (*n* = 3 mice/group; unpaired two-tailed Student's *t* test). **c** Lung Endothelial cells (LEC) isolated using anti-Cd31 antibody from control sgRNA (WT) or FoxO1-sgRNA1 (*FoxO1ΔEC*) mice were used for IB to assess the expression of FoxO1. **d, e** WT and *FoxO1ΔEC* mice were challenged with i.p. LPS (10 mg/kg) or saline for 6 h and lungs were used for IB to determine the expression of FoxO1 (**d**), CHFR (**e**), or VE-cadherin and Ang-2 (**f**). Shown are mean values ± SEM (**d**, *n* = 4 mice/ genotype/group; **e**, *n* = 3 mice/genotype/group; **f**, *n* = 4 mice/genotype/group; one-way ANOVA followed by Tukey's post-hoc test). **g** WT and *FoxO1ΔEC* mice were challenged with LPS as above lungs were used to measure VE-cadherin (VE-cad) ubiquitylation via K48-linked polyubiquitin chains (*n* = 2 independent experiments). Results show a representative blot. **h** WT and *FoxO1ΔEC* mice were challenged with i.p. LPS (10 mg/kg) or saline for 6 h and used to assess lung vascular permeability by measuring EBA uptake. Shown are mean values ± SEM (*n* = 4 mice/genotype/group; two-way ANOVA followed by Tukey's post-hoc test). **i** Survival of WT and *FoxO1ΔEC* mice after CLP (*n* = 8 mice/genotype; WT vs *FoxO1ΔEC*; log-rank test). **j** Model for E3 ligase CHFR regulation of endothelial barrier integrity. Expression of CHFR in ECs downstream of TLR4-NF-κB-FoxO1 axis promotes VE-cadherin degradation via proteasome through K48-linked ubiquitylation of VE-cadherin to induce inflammation. "Created with http://BioRender.com.

drying (1000 RPM for 5 min at RT) and scanned (Typhoon by GE with GFP filters at 10-micron resolution).

## Data analysis
Microarray images were gridded and quantitated using GenePix Pro (v7) software. Microarray images acquired without *h*VE-cad-GFP protein treatment served as negative control. Median intensities (features and local backgrounds) were utilized, and signal-to-noise ratio (SNR) was calculated. Duplicate features (representing identical sample protein) were summarized by average and standard deviation. Although the calculation of standard deviation technically requires three data points, Excel will report a value from two, useful for estimating the reproducibility in signal estimate. Values were then normalized to biological control within the CDH5-treated array before being Loess transformed by print tip and location to remove technical sources of error[52], resulting in the final estimate of magnitude change (*M*-value). *t* test was used to assess the statistical significance (*p*-value) that each *M* value deviated from zero (null hypothesis *M* = 0). A threshold of 95% confidence ($p < 0.05$) was employed to filter data. Gene Ontology (GO) clustering was performed[53] to identify categories (biological processes, cellular components, or molecular functions) overrepresented within this data set relative to what chance alone would predict.

## Cells
Human lung microvascular endothelial cells (HLMVEC; catalog #CC-2527) were obtained from Lonza Walkersville Inc. Wildtype Mouse (C57BL/6) lung endothelial cells (catalog #C57-6011) were from Cell Biologics (Chicago, IL). Both cell types were used between passages 3 and 6. HLMVEC grown to ~80% confluence on gelatin-coated culture dishes were transfected with target siRNAs or Sc-siRNA as described[11]. At 72 h after transfection, cells were used for experiments. The human dermal microvessel endothelial cell line (HMEC) obtained from Dr. Thomas J. Lawley (Emory University School of Medicine, Atlanta, GA) was grown in endothelial basal medium MCDB131 supplemented with 10% FBS, 10 ng/ml epidermal growth factor, 2 mM L-glutamine, and 1 µg/ml hydrocortisone[54].

## Transfections and anti-GFP antibody bead pull-down assay
HMEC grown to near confluency were transfected with plasmids encoding N-terminal eGFP fused WT *h*-CHFR, ΔFHA-*h*CHFR, ΔRF-*h*CHFR, ΔCR-*h*CHFR, or ΔPBZ-*h*CHFR using Lipofectamine-2000[55]. WT *h*-CHFR or mutant CHFRs binding to endogenous VE-cadherin was determined by anti-GFP antibody agarose beads (GFP-Trap Agarose, Chromotek catalog #gta-10) pull down. After 48 h post-transfection, cells were lysed in lysis buffer as described above. Cell lysates were centrifuged (30,000 × *g* for 10 min) to remove insoluble materials. Clear supernatant collected containing 150 µg protein was incubated with 30 µl anti-GFP agarose beads (ChromTech, catalog #gta-10) for 2 h at 4 °C on a rotating platform. Thereafter, the beads were briefly washed 4 times with wash buffer (10 mM Tris-HCl, pH7.5, 150 mM NaCl, 0.1% Triton X-100) by centrifugation at 500 × *g* for 5 min and then used for IB analysis.

## Ubiquitylation assay
HEK293 cells (ATCC, catalog #CRL-1573) were transfected with HA-tagged ubiquitin (HA-Ub) (0.5 µg/ml) alone or cotransfected with plasmids encoding WT-CHFR (1.5 µg/ml), or ΔFHA-CHFR (1.5 µg/ml), or ΔRF-CHFR (1.5 µg/ml) using CalPhos mammalian transfection kit from ClonTech laboratories. After the 24 h post-transfection, cells were infected with recombinant adenovirus expressing C-terminal GFP-tagged *m*VE-cadherin (5 pfu/cell). Twenty-four hours after infection, cells were pretreated with MG132 (10 µM) for 4 h and then cell lysates prepared were immunoprecipitated with pAb against VE-cadherin and blotted with ubiquitin K48-linkage-specific mAb or ubiquitin K63-linkage-specific mAb.

## Transendothelial electrical resistance measurements
Real-time changes in transendothelial monolayer resistance (TER) were measured to assess endothelial barrier integrity[56] using the ECISZθ system (Applied Biophysics, Inc.). HLMVEC transfected with control-siRNA (Sc-siRNA) or CHFR-siRNA (100 nM) were grown to confluency on gold microelectrodes incubated with 2% FBS containing medium for 2 h and then exposed to LPS or PAR-1 activating peptide. Data are presented as resistance normalized to its starting value zero time.

## Mice
All animal experiments were performed under the protocol approved by the office of Animal care and Institutional Biosafety (OACIB) at the University of Illinois at Chicago. Mice were maintained in pathogen-free environment at the University of Illinois Animal Care Facility with 12 h dark and 12 h light cycle, temperature at approximately 23 °C, and humidity at 40-60%. Mice had free access to water and food. Eight to twelve weeks old both male and female mice were used for experiments. Each experiment, the number of mice used were indicated in the figure legends. *Chfrflox/flox* mice were custom made on the C57BL/6 background utilizing a targeting vector flanking exon 3 of mouse *Chfr gene* with 2 loxP sites by Biocytogen (Worcester, MA). We generated endothelial cell (EC)-restricted *Chfr* knockout (*ChfrΔEC*) mice by crossing *Chfrflox/flox* (*Chfrfl/fl*) with B6.Cg-Tg (Cdh5-Cre)7Mlia/J (VE-cadh-Cre+) transgenic mice (Jackson Labs). We also generated EC-restricted FoxO1 knockout (*FoxO1ΔEC*) mice using CRISPR/Cas9 knock-in mice (Rosa26-LSL-Cas9 knock-in on B6j [Rosa26-floxed STOP-Cas9 knock-in on B6J]) (Jackson Labs) as described previously[39]. Here, we first generated CRISPR/Cas9-cdh5-Cre+ mice by crossing CRISPR/Cas9 knock-in mice with Cdh5-Cre transgenic mice (Jackson Labs). Next, we deleted FoxO1 expression in EC (EC-restricted FoxO1 knockout; *FoxO1ΔEC*) mice by liposome-mediated in vivo delivery of the pGS-gRNA plasmid encoding sgRNA-1 (5′-CACGGGGGTCAAGCGGTTCA-3′) or sgRNA-2 (5′-AATTCGGTCATGCCAGCGTA-3′) to target the *mFoxO1* gene in CRISPR/Cas9-cdh5-Cre mice. CRISPR/Cas9-cdh5-Cre mice received pGS-gRNA plasmid

encoding scrambled sgRNA (Sc-sgRNA) (5′-GCGAGGTATTCGGC TCCGCG-3′) considered as wildtype[39]. The liposomes and pGS-gRNA plasmid mixture were injected i.v. into CRISPR/Cas9-cdh5-Cre+ mice (30 µg of plasmid pGS-gRNA plasmid in 100 µl suspension/mouse)[35,57,58]. Liposomes were prepared as previously described by us[35,57,58]. Briefly, the mixture comprised of dimethyldioctadecylammonium bromide and cholesterol (1:1 molar ratio) was dried using the Rotavaporator (Brinkmann) and dissolved in 5% glucose followed by 20 min sonication. The complex consisting of plasmid DNA and liposomes were combined at a ratio of 1 µg of DNA to 8 nmol of liposomes. Four days after sgRNA delivery, mice were used for experiments.

### Vascular injury model in mice
Experimental lung injury was induced by systemic LPS (i.p.) in mice[59]. To assess *Pseudomonas aeruginosa* (PA) infection-induced lung injury, we administered PA (GFP-PA01) intratracheally (i.t.)[60]. Vascular permeability in lung[11,35] and skin[13] was assessed by Evans blue dye bound albumin (EBA) uptake. For histology, paraffin-embedded sections 5 µm in thickness prepared from the lungs were stained with hematoxylin and eosin[59]. For MPO assay, lungs were perfused with PBS or removal of all blood, then were used for MPO activity measurement[35]. Polymicrobial sepsis was induced by the cecal ligation and puncture (CLP) method[59]. The caecum was punctured using 18-gauze needle on 5 different places. For survival studies, mice were monitored four times daily.

### Mouse lung endothelial cells
Lung endothelial cells (LEC) from mice were isolated with mAb to the adhesion molecule CD31 (platelet/endothelial cell adhesion molecule-1 [PECAM-1])[61]. Lungs from 3 mice were minced and digested with 10 ml of collagenase A (1.0 mg/ml; Roche, Catalog #11088785103) in Hank's balanced salt solution (HBSS) for 60 min at 37 °C with gentle shaking. The released cells were centrifuged at 200 x $g$ for 10 min and suspended in 10 ml suspension buffer ($Ca^{2+}$- and $Mg^{2+}$-free PBS containing 0.5 g/100 ml bovine serum albumin, 2 mM EDTA, and 4.5 mg/ml d-glucose), and filtered through 200-µm mesh filter. The filtered cell suspension was centrifuged at 200 x $g$ for 10 min and the cell pellet was suspended in 10 ml of suspension buffer. To this cell suspension, 1.5 µg/ml anti-mouse PECAM-1 antibody (EMD Millipore, catalog # CBL1337) was added and incubated at 4 °C for 30 min with gentle shaking. The cell suspension was centrifuged to remove unbound antibody and washed once with suspension buffer. Then the washed cells were incubated with Dynabeads M-450 (Sheep anti-rat IgG; Invitrogen catalog # 110.35) for 30 min at 4 °C. Thereafter, the cell suspension was attached to a magnetic column and the unbound cells were aspirated. Cells bound to the magnetic beads were washed once with HBSS and digested with trypsin for 3 min at 22 °C. The cells released from the magnetic beads were separated, washed, and suspended in growth medium EBM-2 (Lonza, catalog #CC-3156) supplemented with EGM-2 MV growth factors (Lonza, catalog # CC-4147) and 10% FBS. The cell suspension was plated on Matrigel (BD Bioscience, catalog #354234)-coated 35-mm culture dish and allowed to grow to confluence. Cells were then harvested from the Matrigel plates by dispase (BD Bioscience, catalog #354235) digestion for 60 to 90 min. Cells were washed after dispase treatment once with growth medium and plated on 0.1% gelatin coated culture dish.

### In vivo real-time imaging of PMN transmigration in mouse lungs
We imaged in real-time the transmigration of PMNs across the lung capillaries utilizing 2-photon microscope as we described[36]. *Pseudomonas aeruginosa* (GFP-PA01; $10^7$ CFU/mouse) in 50 µl of PBS was intratracheally injected using micro aerosol spray after retro-orbital injection of Alexa 594-labeled Ly6G antibody (20 µg/mice) (clone 1A8, Biolegend). Lungs were imaged after 6 h of *PA* solution injection. Retro orbital injections of Brilliant Violet 421-labeled LY6G antibody

(10 µg/mice) (clone 1A8, Biolegend) and SeTau647-labeled CD31 antibody (20 µg/mice) (clone 390, Biolegend) were made to stain PMNs and lung vessels, respectively, before the surgery. A resonant-scanning two-photon microscope (Ultima Multiphoton Microscopes, Bruker) with an Olympus XLUMPlanFL N 20x (NA 1.00) was used to collect tetra-color images (Emission filter; 450/70 nm for Brilliant Violet 421, 525/50 nm for GFP and 595/50 for Alexa 594 and 708/75) with 820 nm excitation. Two-photon microscope images were processed and analyzed using the following softwares:

Origin (https://www.originlab.com/index.aspx?go=Products/Origin);

ImageJ (https://imagej.net/software/fiji/downloads)

LabVIEW (https://www.ni.com/en-us/shop/labview.html?cid=Paid_Search-7013q000001UgkyAAC-Consideration-GoogleSearch_102713974313&s_kwcid=AL!6304!3!449107487685!e!!g!!national%20instruments%20labview&gclid=CjwKCAjw-8qVBhANEiwAfjXLrvSpZCKoowgJg5HoP9_vU0LXoZOT0ccfOuG-mG7n3itBQq-30hAHGxoCtNcQAvD_BwE;

CoVSTii (https://github.com/YoshiTsukasaki/CoVSTii) (https://uofi.box.com/v/Stand-aloneCoVSTii)

### Human lung specimens
Clinically evaluated non-surviving Sepsis/ARDS and control patient lung specimens were collected during autopsy at VU Medical Center, Amsterdam, the Netherlands under the approved Institutional protocol (protocol number BUP2020-62). Deidentified paraffin-embedded lung tissue sections were used for immunostaining.

### Immunostaining of endothelial cells
HLMVEC grown to confluency on glass coverslips were washed quickly with ice-cold phosphate-buffered saline (PBS) and fixed with 2% PFA for 15 min at 4 °C. Following fixation, cells were permeabilized with 0.05% Triton X-100 for 1 min at 4 °C. Next, cells were washed three times with PBS and then incubated with blocking buffer (PBS containing 5% horse serum and 1% BSA) for 1 h, at RT. Cells were then incubated overnight with the indicated primary antibody (in PBS containing 1% BSA) at 4 °C. The next day, cells were washed three times and incubated with specific Alexa-Fluor conjugated secondary antibody and DAPI for 1 h at 4 °C. Finally, cells were washed three times and mounted on glass slides for viewing. Images were acquired with the Zeiss LSM 880 confocal microscope.

### Immunostaining of lung tissue sections
Briefly paraffin embedded formalin fixed sections were deparaffinized by passing through a xylene and ethanol series followed by rehydration in $dH_2O$[35]. The antigen retrieval was done by boiling the slides in antigen retrieval buffer (10 mM citrate, 0.05% Tween 20, pH 6.0) for 20 min. Next the sections were incubated in a blocking buffer (PBS containing 4% BSA, 0.2% Triton X-100) for 2 h RT. After this, the sections were incubated with indicated primary antibodies overnight in 0.5% BSA blocking buffer and staining was detected by incubation with the appropriate fluorochrome coupled secondary antibody for 1 h RT. The sections were mounted with a DAPI-supplemented anti-fade mounting solution (Invitrogen) and the images were acquired with the Zeiss LSM 880 confocal microscope.

### Immunoprecipitation and immunoblotting
HLMVEC grown to confluence treated with or without specific agents were three times washed with phosphate-buffered saline (PBS) at 4 °C and lysed in lysis buffer. Mouse lungs were homogenized in lysis buffer[11]. Endothelial cell lysates or mouse lung homogenates were centrifuged (30,000 × $g$ for 10 min) to remove insoluble materials. Clear supernatant collected 50 µg protein was subjected to immunoblot analysis. For immunoprecipitation, clear supernatant 300 µg of protein was used. Each sample was incubated overnight with

indicated antibody at 4 °C. Next day, Protein A/G beads were added to the sample and incubated at 4 °C for 1 h. Protein A/G beads were then washed three times with wash buffer (Tris-buffered saline containing 0.05% Triton X-100, 1 mM $Na_3VO_4$, 1 mM NaF, 2 μg/ml leupeptin, 2 μg/ml pepstatin A, 2 μg/ml aprotinin, and 44 μg/ml phenylmethylsulfonyl fluoride) and used for immunoblot analysis. Endothelial cell lysates, lung tissue extracts, or immunoprecipitated proteins were boiled in Laemmli sample buffer (50 mM Tris-HCl pH 6.8, 2% SDS, 10% glycerol, 0.05% bromophenol blue, and 2.5% β-mercaptoethanol) and resolved by SDS-PAGE on a 4-15% gradient or 10% separating gel and transferred to a polyvinylidene difluoride (PVDF) membrane. For the detection of ubiquitylation of proteins, immunoprecipitated proteins were boiled in sample buffer (100 mM Tris-HCl pH 6.8, 4% SDS, 20% glycerol, 0.2% bromophenol blue, and 200 mM DTT) and then proteins were resolved by SDS-PAGE on a 6% separating gel. Membranes were blocked with 5% dry milk in TBST (10 mM Tris-HCl pH7.5, 150 nM NaCl, and 0.05% Tween-20) at RT for 1 h. Membranes were then probed with the indicated primary antibody (diluted in blocking buffer) overnight at 4 °C. Next, membranes were washed 3 times and then incubated with appropriate HRP-conjugated secondary antibody. Protein bands were detected by enhanced chemiluminescence. Immunoblot bands were quantified using NIH ImageJ software.

## Quantitative Real-Time PCR

Total RNA from mouse lung tissue and mouse lung endothelial cells were isolated by RNeasy Plus kit (Qiagen,catalog #74034). Concentrations of isolated RNA samples were measured using Nanodrop 1000 (Thermo Fisher) at 260 nm, and purity was assessed by 260/280 nm optical density ratio. RNA samples were transcribed to complementary DNAs with oligo(dT) primers using "RevertAid First Strand cDNA Synthesis Kit" (Thermo Fischer, catalog #K1621). Quantitative gene expressions were determined by real-time PCR upon mixing the cDNAs with Fast SYBR Green Master Mix (Applied Biosystems, catalog #4385612) with specific qPCR primers. The real-time amplification was carried out using Quant studio7 Flex Real-Time PCR System (Applied Biosystems). Results were calculated using the comparative CT-method (60) and expressed relative to the expression of the housekeeping gene. The following mouse primers were used: *Chfr* (forward: 5′-TCTGGTGAGGTGACACTGGA-3′, reverse: 5′-GCCACATTGTGTTCTG GCTC-3′), FoxO1 (forward:5′-CAAGGCCATCGAGAGCTCAG-3′, reverse: 5′-AATTGAATTCTTCCAGCCCGC-3′), ICAM1 (forward 5′-AAAGATCAC ATGGGTCGAGG-3′, reverse 5′-AAAGTAGGGGAGGTGGTGCT-3′), E-selectin: (forward 5′-TAGCGCCTGGATGAAAGCAA-3′, reverse 5′-GAGC TCACTGGAGGCATTGT-3′), P-selectin: (forward 5′-GCTGTTGCTGAA AACGGGAG-3′, reverse 5′-TGCCCACCAACATTGCTACT-3′), MMP3: (forward, 5′-CAGTCCCTCTATGGAACTCCC-3′, reverse, 5′-AGGGTGC TGACTGCATCAAA-3′), MMP9: forward 5′-CTCTCCTGGCTTTCGGCT G-3′, reverse 5′-AGCGGTACAAGTCCTCTGC-3′), GAPDH: (forward, 5′-A CCCAGAAGACTGTGGATGG-3′, reverse, 5′-CACATTGGGGGTAGGAAC AC-3′), β-actin (forward:5′-ACTCCTATGTGGGTGACGAG-3′, reverse, 5′-ATCTTTTCACGGTTGGCCTTAG-3′).

## Promoter analysis and ChIP assay

Consensus binding sites for transcription factors in 5′-regulatory regions of mouse and human genes were analyzed with the Eukaryotic promoter Database (SIB; https://epd.epfl.ch//index.php). ChIP assays were performed as we described previously[59] using ChIP assay kit from Millipore (catalog #17-295). The primers used for PCR after ChIP to determine FoxO1 binding sites in *mChfr* gene: region −457 to −107, forward 5′-ATCAGCCCTCAGAGACGAGT-3′, reverse 5′-TAGAGCGG TGCCAAAAGG-3′; region −651 to −258, forward 5′-CCGAGCTGCTC AAGAACCA-3′, reverse 5′-GCACCATGTTAGATTCGGGC-3′; region −1874 to −1563, forward 5′-ACAGTTTCCTGACTAGTGTGCC-3′, reverse 5′-ACACAATTGGTTTCCAAGGGTATAG-3′.

## Statistical analysis

Student's *t* test, ANOVA followed by Tukey's or Bonferroni's post-hoc test, and log-rank test were used to determine statistical significance. For all bar graphs, data are represented as mean values ± SEM. *p*-values < 0.05 were considered significant. All calculations were performed using GraphPad Prism 9 software.

## Data availability

All data needed to reproduce the results are available within the manuscript supplementary information and source data provided with this paper. Other relevant data is available from the corresponding authors. Source data are provided with this paper.

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

## Acknowledgements
This work was funded by NIH grants: R01GM138499 (C.T.), R01HL156965 (C.T.), and P01HL160469-project 2 (A.B.M). The schematic diagrams were made using Biorender.com.

## Author contributions
C.T.: fund acquisition, conceived, study design, experimental work, data interpretation, supervision, and manuscript writing; D.M.W.: mouse models development, experimental work, and data collection; M.O.A.: experimental work and data collection; S.B.: experimental work and data collection; Y.T.: performed intravital imaging; A.M.: experimental work; J.C.J.: experimental work; C.L.: performed protein microarray study; H.W.M.N.: obtained human lung tissues; A.B.M.: fund acquisition, study design, and manuscript writing.

## Competing interests
The authors declare no competing interests.
