## [Peer Review File · Nature Communications]

REVIEWER COMMENTS

Reviewer #1 (Remarks to the Author):

In the current manuscript, Tiruppathi et al describe the role of ubiquitin E3 ligase, Checkpoint protein with FHA and Ring domain (CHFR), as a degrading signal for VE-cadherin through K48-linked ubiquitylation at K633/K690 in lung microvascular endothelial cells. EC-specific CHFR-KO mice exhibited augmented expression of VE-Cadherin and resisted LPS- and pseudomonas-induced barrier breakdown, and inflammatory cell infiltration in the lung and brain. The effect was reliant on FoxO1 downstream of the LPS receptor, TLR4.

The team has done an extensive study in multiple cellular, biochemical, and animal models demonstrating that preventing VE-Cadherin degradation via knocking out CHFR in ECs is sufficient to prevent LPS- and bacteria-induced vascular leakage in the lung and brain, and infiltration of neutrophils.

Although the presented data is convincing, the evidence is limited to tissue sections. No analysis of broncho-alveolar lavage (BAL) samples was performed. Analysis of BAL samples from mice for protein content, number of infiltrated neutrophils, and MPO activity would strengthen the conclusions.

TLR4 is highly expressed in inflammatory cells, including macrophages and neutrophils. Activation of these cells in a sepsis model is not reliant on the expression levels of VE-Cadherin on ECs. How do the authors explain a complete inhibition of inflammatory cell infiltration in CHFR-KO mice? Does CHFR have effects on molecules other than VE-Cadherin, which also might contribute to the phenotype? An RNA sequencing or micro-array analysis would help to provide more clarity on the VE-Cadherin independent function of CHFR in the regulation of the EC barrier.

No analysis of tight junction proteins is performed. Although FoxO has been demonstrated to modulate EC claudin-5 expression, Cldn5 KO mice have no vascular leakage. A number of other molecules such as several MMPs have also been shown to be regulated by Foxo1 in the modulation of vascular permeability in the lungs. Expression analysis of other proteins might provide additional insights supporting the current findings that CHFR indeed is a vascular permeability-promoting molecule.

VEGF, FGF, and Angiopoietin-1, etc. also modulate FoxO transcriptional activity. However, the effect of only bacteria and LPS are studied in the manuscript. What effects do physiological inducers of vascular permeability such as VEGF, Thrombin, TNF, and/or histamine, etc. have on EC-barrier leakage in vitro, and edema/Miles assay in vivo?

CHFR is known to regulate the cell cycle by delaying entry into the metaphase. Do CHFR-deficient ECs hyper-proliferate?

Overall, this is a very interesting study with some novel information. Identifying other signaling molecules modulated in mice deficient in CHFR in ECs, which are signaling partners of VE-Cadherin pathway will strengthen the conclusion of the study.

Reviewer #2 (Remarks to the Author):

This manuscript investigates the role of the E3 ligase CHFR for ubiquitylation of VE-cadherin and the barrier function of the blood vessel wall. It is shown that CHFR gene inactivation in endothelial cells impairs baseline as well as LPS-induced VE-cadherin ubiquitylation, enhances VE-cadherin expression and inhibits LPS induced vascular permeability in vitro and in vivo. Likewise, CHFR deficiency reduced a bacterial infection-induced lung vascular barrier breakdown, PMN recruitment and mortality. In addition, binding of the transcription factor FoxO1 to the mChfr promotor was found to be induced in endothelial cells upon LPS stimulation, and LPS induced expression of the CHFR protein was inhibited by a FoxO1 inhibitor, and FoxO1-sgRNA driven gene inactivation (in mice expressing the CRISPR/cas9 system in endothelial cells). Likewise, FoxO1 deletion reduced LPS induced VE-cadherin degradation in the lung, VE-cadherin ubiquitylation, and vascular permeability. In addition, survival in a cecal ligation and puncture model was improved.

The identification of CHFR as regulator of vascular permeability is indeed interesting. However, a role of FoxO1 in LPS-induced vascular barrier break down in mice via effects on VE-cadherin expression has been published before – a study which is not cited. The published study (Zhao et al., *Ann. Transl. Med.* 2022, Vol 10, No 11 (June 2022) :630) showed that LPS induced expression of FoxO1 contributes to enhanced vascular permeability and loss of VE-cadherin and ZO-1 via an autophagy mediated mechanism. The present study explains the role of FoxO1 by inducing CHFR and VE-cadherin degradation.

The following points should be addressed:

- 1) The paper by Zhao et al., Upregulation of FoxO1 contributes to LPS-induced pulmonary endothelial injury by induction of autophagy, *Ann. Transl. Med.* 2022, 10(11):630, should be cited as well as another paper that shows that FoxO1 mediates LPS-induced lung edema in mice (Artham et al., *Pharmacol. Res.*, 2019, 141:249-263). It should be discussed, that the contribution of FoxO1 to LPS-induced lung edema formation is not fully explained by the induction of CHFR.

2) Extended data Fig. 1g: The co-localization data of VE-cadherin and different forms of CHFR show intracellular structures which are undefined. It should be made clear what these structures are and the resolution of these images should be improved.

3) Evidence for the binding of CHFR to the cytoplasmic C-terminus of VE-cadherin is only based on the microarray screen, performed by the proteomics company. It should be tested whether this binding can be verified in endothelial cells by co-immunoprecipitations. Relevance of the enzyme for VE-cadherin ubiquitylation is based on overexpression studies and siRNA experiments with ubiquitylation as readout. This does not exclude that CHFR may affect VE-cadherin indirectly.

4) Evidence for the relevance of K633 and K690 for VE-cadherin degradation is only based on loading endothelial cells with unknown quantities of model peptides containing these respective K residues. This is rather indirect evidence. Are these lysine residues indeed relevant within the sequence of VE-cadherin for LPS-induced degradation? In addition, how was VE-cadherin staining quantitated (Fig. 1i), since some cells especially in the bottom row of micrographs show very strong nuclear background staining.

5) There is no method described in this manuscript for isolating mouse lung endothelial cells.

6) The in vivo effect of FoxO1-sgRNA1 is only based on a single sgRNA. How can it be excluded that off target effects occurred?

7) The deletion mutants of CHFR (ΔFHA and ΔRF) are not described in detail.

8) Discuss the contradicting results of K48-linked and K63-linked ubiquitylation of VE-cadherin (Dejana group versus this group)

9) The manuscript shows, that LPS induces the loss of the VE-cadherin blot signal. However, the blots were performed with polyclonal antibodies against a C-terminal peptide of VE-cadherin. Thus, cleavage of the very C-terminus (beyond aa 750) would be sufficient to lose the blot signal. How would these blots look with antibodies against the N-terminus of VE-cadherin? In addition, it should be shown, whether these effects are indeed based on endocytosis.

Reviewer #3 (Remarks to the Author):

This Ms studies how endothelial E3 ligase CHFR functions as a crucial regulator of endothelial integrity and inflammation, and controls leukocyte extravasation as well as LPS-induced permeability. Mechanistically, they show that CHFR works through VE-cadherin and is involved the internalization and ubiquitination of VE-cadherin.

Lots of the data are gathered from in vivo experiments and show solid results. Some general remarks: to avoid any potential side effects, I would recommend adding a second siRNA in fig 1.

Also, to test if CHFR is specifically targeting VE, I would recommend repeating some of the functional assays where CHFR is reduced in protein in a VE-cad-deficient background, i.e., knock down of VE. These control experiments are important to strengthen the message and point out specificity.

1. Ext data fig 1: add kDa along ubi blot and VE and CHFR co-localize, not associate. That conclusion cannot be drawn from IFs. Please add some arrows for clarity in the magnification.
2. When VE expression is down upon LPS treatment, is also beta-cat/alpha-cat down or is Ve replaced by N-cadherin?
3. The authors state, "...E3 ligase CHFR primarily attaches K48-linked poly-Ub chains to VE-cadherin...". Can the authors provide information that reveals where on the intracellular tail of VE the Ubs are attached? Is that near the beta-catenin or p120 catenin binding sites?
4. Data shown in fig 2d are interesting: VE-PTP is also increased. Is VE-PTP also regulated by CHFR and is this needed to start VE-internalization. Literature shows that p120 regulates VE internalization. How does that integrate with the VE-Ubi and degradation in this study. Is VE-p120 interaction increased upon CHFR knockdown?
5. It is clear that the depletion of CHFR increases the levels of VE-cad. However, it is not clear if these levels are indeed increased on the cell surface. Only showing Ifs with VE staining is not enough. To show this properly, they need to do an antibody-feeding experiment.
6. For the inflammation data, they need to address the expression levels of at least some of the essential molecules, like selectins and ICAM-1 to really attribute the effects to CHFR-mediated increase in VE expression.

Reviewer #4 (Remarks to the Author):

In this manuscript, Tiruppathi and colleagues have identified CHFR as an ubiquitin E3 ligase that targets VE-cadherin for ubiquitination and degradation. The authors have used AVM BIOMED Snapshot Proteomics Microarray system and have identified CHFR as a VE-cadherin binding protein. The authors have generated an EC-specific CHFR KO mice and show that EC deficient CHFR mice are resistant for TLR4-induced VE-cadherin degradation. Although, this paper is an interesting work, however several conceptual and technical issues reduced my enthusiasm for the manuscript. First, the authors have used AVM BIOMED Snapshot Proteomics Microarray system to identify CHFR as a VE-cadherin binding protein. The authors provide a scant description about this assay. I even could not find any information about this array in the company's website. From the data provided, I am not clear what regulates the binding of CHFR with VE-cadherin. Additional binding experiments using recombinantly expressed proteins are needed. In vitro ubiquitination assay is required to establish the direct role of CHFR in the ubiquitination of VE-cadherin. Second, CHFR is not known to be present at the plasma membrane where the VE-cadherin is present (CHFR is known to present in the nucleus) this raises a serious question whether CHFR is a physiological regulator of VE-cadherin. Extensive co-localization experiments are needed to address this issue. Third, inactivation of VE-cadherin in mice is lethal with severe impairment in angiogenesis, what is the effect of loss of CHFR in angiogenesis. In addition, the authors need to carry out both in vivo and in vitro vascular permeability assays.

Reviewer #5 (Remarks to the Author):

In this study, Tiruppathi et al use a protein microarray to scout for binding partners of VE-cadherin. The goal of this was to identify ubiquitin ligases that regulate VE-cadherin levels. CHFR was identified as one of the hits and biochemical and cellular validation experiments were carried out to demonstrate that CHFR mediates K48-linked ubiquitination and consecutive degradation of VE-cadherin. Experiments in primary human lung microvascular endothelial cells revealed that LPS induced degradation of VE-cadherin was prevented by siRNA mediated knockdown of CHFR. In the second part of the manuscript, the authors show that in lung endothelial cells of CHFR knockout in mice VE-cadherin levels are increased and this suppresses LPS-induced vascular injury and mortality. Finally, the authors identify FoxO1 as the responsible transcription factor driving the expression of CHFR after TLR4 agonist stimulation.

The manuscript is written clearly but needs revision for language in some parts. The plots in the figures are not always informative and controls are missing for many experiments (see my comments below). I also find small inconsistencies in some experiments.

The methods used need to be described in much more detail (e.g., IP buffer recipe not described, washing of the IPs, affinity purification of peptides for LC-MS etc.).

Comments:

- AVM BIOMED Snapshot Proteomics Microarray system: explain in more detail (e.g., what negative control was used in the assay and how was the significance assessed?). It is not clear to me why the

authors decided to focus on CHFR? It has the highest fold-change in the microarray but also the highest p-value among the 20 proteins shown in Extended Data Figure 1b. The same is true for E3 ligases shown in Extended Data Figure 1c. The authors should present a global view of the screening data, for example in form of a volcano plot, and include it in the main figure.

- Extended Data Figure 1f: Are the expression levels of GFP-CHFR (wt, Δ FHA, Δ RF) comparable? A control blot for CHFR is missing. As additional control, could the authors show that any other of the top-scoring E3s shown in ED Figure 1c does not induce K48-linked polyubiquitylation of VE-cadherin?
- Extended Data Figure 1g: Does not look very convincing to me. Would benefit from a proper quantification.
- Figure 1b: Show WB for CHFR to demonstrate efficient knockdown. Since TLR4 stimulation results in upregulation of CHFR transcription (through FoxO1), I would expect to see a stabilization of CHFR in this experiment (at least at 24 h of stimulation). In general, all experiments that assess either knockdown/knockout of CHFR or inhibition/depletion of FoxO1, with or without LPS stimulation should include blots for VE-cadherin and CHFR.
- Figure 1e: Could the authors comment on the fact that MG-132 treatment does not rescue LPS-induced degradation of VE-cad (see immunoblot for VE-Cad)? On top of showing the immunoblot for VE-Cad, I would also recommend to re-probe for VE-cad using the membrane that was utilized for K48 ubiquitin detection. Also, the control blot showing efficient CHFR knockdown is missing. The same is true for figures 2f and g.
- The authors immunoprecipitate VE-cad, followed by enrichment of diGly remnant peptides to investigate whether LPS stimulation induced VE-cad ubiquitination. Some more details on the setup should be described in the figure legend. For example, that the experiment was done in the presence of MG-132 and that the treatment time was 6 h. I am wondering why the authors chose a double IP approach (i.e., IP VE-cad + IP diGly), instead of directly enriching for K-GG peptides after LPS treatment? Instead of showing MS/MS spectra in the main figure (which are not very informative), the authors should present quantitative results (i.e., levels of all quantified K-GG peptides in control vs treatment vs the corresponding levels of total protein). The authors claim that CHFR attaches K48-linked ubiquitin to K633 and K690 of VE-cad; however, diGly remnant profiling does not allow determining the ubiquitin linkage type. This statement should be removed or else an orthogonal, MS-based approach (e.g., TUBEs (tandem ubiquitin binding entities)) should be employed.
- A negative control should be included for the experiment shown in figure 1i (for example, VE-pep-1 in which the lysine is substituted by an arginine)

Response to reviewers' comments:

Reviewer#1:

“In the current manuscript, Tirupathi et al describe the role of ubiquitin E3 ligase, Checkpoint protein with FHA and Ring domain (CHFR), as a degrading signal for VE-cadherin through K48-linked ubiquitylation at K633/K690 in lung microvascular endothelial cells. EC-specific CHFR-KO mice exhibited augmented expression of VE-Cadherin and resisted LPS- and pseudomonas-induced barrier breakdown, and inflammatory cell infiltration in the lung and brain. The effect was reliant on FoxO1 downstream of the LPS receptor, TLR4.

The team has done an extensive study in multiple cellular, biochemical, and animal models demonstrating that preventing VE-Cadherin degradation via knocking out CHFR in ECs is sufficient to prevent LPS- and bacteria-induced vascular leakage in the lung and brain, and infiltration of neutrophils.”

Response: We appreciate the reviewer's positive comments of our manuscript.

“Although the presented data is convincing, the evidence is limited to tissue sections. No analysis of broncho-alveolar lavage (BAL) samples was performed. Analysis of BAL samples from mice for protein content, number of infiltrated neutrophils, and MPO activity would strengthen the conclusions.”

Response: In response, we performed additional *in vivo* experiments utilizing WT and Chfr-EC-KO (*Chfr^{ΔEC}*) mice and now have included the BAL sample analysis data (Figure #4b). Here we observed substantially reduced protein content, number of infiltrated neutrophils, and MPO activity in the BAL fluid of LPS challenged *Chfr^{ΔEC}* mice compared with WT mice (Figure #4b).

“TLR4 is highly expressed in inflammatory cells, including macrophages and neutrophils. Activation of these cells in a sepsis model is not reliant on the expression levels of VE-Cadherin on ECs. How do the authors explain a complete inhibition of inflammatory cell infiltration in CHFR-KO mice?”

Response: Published studies from the Vestweber group (ref # 2, 13) utilizing genetically modified mouse models have shown that stabilizing VE-cadherin at endothelial adherens junctions (AJs) blocked LPS-induced PMN extravasation and vascular permeability. We observed that more than 3-fold increased expression of VE-cadherin and its interacting proteins (VE-PTP, p120-catenin, and β -catenin) in *Chfr^{ΔEC}* mice compared with WT counterparts (Figure # 3c, d, e); therefore, it is possible that increased expression of VE-cadherin and its interacting proteins could potentially increase the strength of the endothelial barrier. Thus, based on our findings the markedly reduced in PMN extravasation and vascular permeability in *Chfr^{ΔEC}* mice is due to strength of adhesive interactions of endothelial AJs.

“Does CHFR have effects on molecules other than VE-Cadherin, which also might contribute to the phenotype? An RNA sequencing or micro-array analysis would help to provide more clarity on the VE-Cadherin independent function of CHFR in the regulation of the EC barrier.”

Response: Yes, we agree that RNA sequencing helps to provide clarity concerning the VE-cadherin independent function of CHFR in regulating the EC barrier. However, in the present studies, we focused on the role of CHFR in mediating ubiquitylation of VE-cadherin and its effect on endothelial barrier integrity. The results are already comprehensive and convincing with the many changes made in response to the previous, we will certainly carry out RNA sequencing in future studies.

“No analysis of tight junction proteins is performed. Although FoxO has been demonstrated to modulate EC claudin-5 expression, Cldn5 KO mice have no vascular leakage. A number of other molecules such as several MMPs have also been shown to be regulated by Foxo1 in the modulation of vascular permeability in the lungs. Expression analysis of other proteins might provide additional insights supporting the current findings that CHFR indeed is a vascular permeability-promoting molecule.”

Response: In response, we showed that the augmented expression of claudin-5 (tight junction protein) and VE-cadherin interacting proteins (VE-PTP, p120-catenin, and β -catenin) in *Chfr*^{ΔEC} mice (Figure #3e). Further, in *Chfr* deficient endothelial cells, we observed substantially reduced expression of the FoxO1 target MMP3 (Supplementary Fig. 4a, b). Furthermore, we did not observe any significant difference in the expression of MMP9 and PMN adhesive molecules ICAM-1, E-selectin and P-selectin between WT and *Chfr*^{ΔEC} mice (Supplementary Fig. 4a, b).

“VEGF, FGF, and Angiopoietin-1, etc. also modulate FoxO transcriptional activity. However, the effect of only bacteria and LPS are studied in the manuscript. What effects do physiological inducers of vascular permeability such as VEGF, Thrombin, TNF, and/or histamine, etc. have on EC-barrier leakage in vitro, and edema/Miles assay in vivo?”

Response: As suggested, we have added new *in vivo* vascular permeability measurements to support the claim that CHFR is essential for degrading VE-cadherin in endothelial cells. We showed that PAR-1 (thrombin receptor)-, TNF- α -, and VEGF-induced increases in vascular permeability were blocked in *Chfr*^{ΔEC} mice compared with WT (Supplementary Fig. 3). In addition, *in vitro* thrombin receptor PAR-1-induced permeability increase was also blocked in CHFR knockdown endothelial cells (Figure # 2f).

“CHFR is known to regulate the cell cycle by delaying entry into the metaphase. Do CHFR-deficient ECs hyper-proliferate?”

Response: We now include data showing that CHFR deficient EC hyper-proliferate as compared with control ECs (Supplementary Fig. 5)

“Overall, this is a very interesting study with some novel information. Identifying other signaling molecules modulated in mice deficient in CHFR in ECs, which are signaling partners of VE-Cadherin pathway will strengthen the conclusion of the study.”

Response: We thank the reviewer for the positive comments about our paper. We have revised the manuscript with several additional experiments to support our conclusion that endothelial expressed CHFR is a crucial regulator of endothelial barrier integrity and vascular inflammation.

Reviewer #2:

“This manuscript investigates the role of the E3 ligase CHFR for ubiquitylation of VE-cadherin and the barrier function of the blood vessel wall. It is shown that CHFR gene inactivation in endothelial cells impairs baseline as well as LPS-induced VE-cadherin ubiquitylation, enhances VE-cadherin expression and inhibits LPS induced vascular permeability in vitro and in vivo. Likewise, CHFR deficiency reduced a bacterial infection-induced lung vascular barrier breakdown, PMN recruitment and mortality. In addition, binding of the transcription factor FoxO1 to the mChfr promotor was found to be induced in endothelial cells upon LPS stimulation, and LPS induced expression of the CHFR protein was inhibited by a FoxO1 inhibitor, and FoxO1-sgRNA driven gene inactivation (in mice expressing the CRISPR/cas9

system in endothelial cells). Likewise, FoxO1 deletion reduced LPS induced VE-cadherin degradation in the lung, VE-cadherin ubiquitylation, and vascular permeability. In addition, survival in a cecal ligation and puncture model was improved. “

Response: We thank the reviewer for correctly summarizing the findings.

“The identification of CHFR as regulator of vascular permeability is indeed interesting. However, a role of FoxO1 in LPS-induced vascular barrier break down in mice via effects on VE-cadherin expression has been published before – a study which is not cited. The published study (Zhao et al., Ann. Transl. Med. 2022, Vol 10, No 11 (June 2022) :630) showed that LPS induced expression of FoxO1 contributes to enhanced vascular permeability and loss of VE-cadherin and ZO-1 via an autophagy mediated mechanism. The present study explains the role of FoxO1 by inducing CHFR and VE-cadherin degradation.”

Response: Thank you for pointing out the recent publications relevant to our studies. These are incorporated in the revision.

“The following points should be addressed: 1) The paper by Zhao et al., Upregulation of FoxO1 contributes to LPS-induced pulmonary endothelial injury by induction of autophagy, Ann. Transl. Med. 2022, 10(11):630, should be cited as well as another paper that shows that FoxO1 mediates LPS-induced lung edema in mice (Artham et al., Pharmacol. Res., 2019, 141:249-263). It should be discussed that the contribution of FoxO1 to LPS-induced lung edema formation is not fully explained by the induction of CHFR.”

Response: In the revision, we have discussed these two recent publications. In addition, we provide several lines of evidence to support the claim that endothelial CHFR plays a crucial role in regulating endothelial barrier integrity and vascular inflammation.

“2) Extended data Fig. 1g: The co-localization data of VE-cadherin and different forms of CHFR show intracellular structures which are undefined. It should be made clear what these structures are and the resolution of these images should be improved.”

Response: We performed new experiments; the results clearly demonstrated the cellular distribution and colocalization of CHFR with VE-cadherin in ECs (Figure # 2g).

“3) Evidence for the binding of CHFR to the cytoplasmic C-terminus of VE-cadherin is only based on the microarray screen, performed by the proteomics company. It should be tested whether this binding can be verified in endothelial cells by co-immunoprecipitations. Relevance of the enzyme for VE-cadherin ubiquitylation is based on overexpression studies and siRNA experiments with ubiquitylation as readout. This does not exclude that CHFR may affect VE-cadherin indirectly.”

Response: We now include co-immunoprecipitation experiments (Figure # 1f, 2c, 3g), which clearly show the interaction between CHFR and VE-cadherin. In addition, we performed *in vitro* ubiquitylation assay. The results show that E3-ligase CHFR directly ubiquitylates recombinantly expressed VE-cadherin (Figure # 1h).

4) Evidence for the relevance of K633 and K690 for VE-cadherin degradation is only based on loading endothelial cells with unknown quantities of model peptides containing these respective K residues. This is rather indirect evidence. Are these lysine residues indeed relevant within the sequence of VE-cadherin for LPS-induced degradation? In addition, how was VE-cadherin staining quantitated (Fig. 1i), since some cells especially in the bottom row of micrographs show very strong nuclear background staining.

Response: We employed LC-MS to determine the critical lysine residues in the C-terminus of VE-cadherin ubiquitylated in endothelial cells. We found that the ubiquitylation occurred at the C-terminal K-633 and K-690 of VE-cadherin (Figure # 2i-l). To address the functional relevance of ubiquitylation sites, we used a cell permeable peptide approach (Ref # 11, 32) to block VE-cadherin ubiquitylation and degradation. We studied the effect of known quantities of cell-permeable VE-cadherin K-633 (Peptide 1) and K690 (Peptide 2) containing peptides on VE-cadherin expression at cell-cell junction with or without LPS challenge (Figure # 2k). We found that these two peptides were essential for ubiquitylation of these sites on VE-cadherin. In addition, as suggested by Reviewer #5, we studied the effects of K/R mutant peptides as a negative control on LPS-induced degradation of VE-cadherin. The K/R mutant peptides had no significant effect on LPS-induced degradation of VE-cadherin (Extended Data Figure # 1d). These results together show the functional relevance of ubiquitylation-mediated degradation of VE-cadherin.

We also quantified the confocal images at cell-cell junctions to assess VE-cadherin expression (Fig. 2k) which is clearly described in Methods. Our results (Figure # 2k) show that VE-cadherin derived cell permeable peptides (peptide 1 and peptide 2) together effectively prevented LPS-induced degradation of VE-cadherin at cell-cell junctions.

“5) *There is no method described in this manuscript for isolating mouse lung endothelial cells.*”

Response: In response, we now include mouse lung endothelial cell isolation protocol in the Methods section.

“6) *The in vivo effect of FoxO1-sgRNA1 is only based on a single sgRNA. How can it be excluded that off target effects occurred?*”

Response: We appreciate the reviewer’s point. We used the CRISPRtool (<http://crispr.mit.edu>) to design the sgRNAs to target mFoxO1 to minimize potential off targets as described in (Ref # 49). We tested two sgRNAs specific to mFoxO1 with very low off-target score (Figure # 6b, c). We observed that sgRNA1 effectively suppressed FoxO1 expression as compared to sgRNA2 (Figure # 6b, c). Based on these results, we used sgRNA1 for further experiments. In addition, we validated our results by determining the expression of the known FoxO1 target gene angiotensin-2 (Ang-2) (Figure # 6f).

“7) *The deletion mutants of CHFR (Δ FHA and Δ RF) are not described in detail.*”

Response: In the revision, we describe CHFR expression constructs preparation in the Methods section.

“8) *Discuss the contradicting results of K48-linked and K63-linked ubiquitylation of VE-cadherin Dejana group versus this group*”

Response: We have discussed the contradicting results in the revised manuscript (Page # 12 & 13). Dejana’s group (Ref # 14) studied VE-cadherin phosphorylation-mediated internalization and ubiquitylation after challenging HUVECs with bradykinin. By immunostaining, they observed that the internalized VE-cadherin was co-localized with K⁶³-linked ubiquitin. In contrast, to define the biochemical mechanisms by which the loss of VE-cadherin expression from AJs occurs in clinical conditions such as acute lung injury and sepsis, we studied LPS-induced effect on VE-cadherin expression in endothelial cells. Here we observed that LPS induced K⁴⁸-linked polyubiquitylation of VE-cadherin (Figure # 2c, 3g, 6g), which in turn triggered degradation of VE-cadherin in endothelial cells.

“9) *The manuscript shows, that LPS induces the loss of the VE-cadherin blot signal. However, the blots*

were performed with polyclonal antibodies against a C-terminal peptide of VE-cadherin. Thus, cleavage of the very C-terminus (beyond aa 750) would be sufficient to lose the blot signal. How would these blots look with antibodies against the N-terminus of VE-cadherin? In addition, it should be shown, whether these effects are indeed based on endocytosis.

Response: As suggested, we used antibodies against N-terminus and C-terminus of VE-cadherin for Western analysis. We failed to detect low molecular weight protein bands (i.e., VE-cadherin cleaved products) using antibodies against either N-terminus or C-terminus of VE-cadherin in LPS challenged cells (Supplementary Fig. 1a). In addition, using confocal imaging, we observed importantly the co-localization of CHFR with VE-cadherin in endocytosed vesicles after LPS challenge (Figure # 2g).

Reviewer #3:

“This Ms studies how endothelial E3 ligase CHFR functions as a crucial regulator of endothelial integrity and inflammation, and controls leukocyte extravasation as well as LPS-induced permeability. Mechanistically, they show that CHFR works through VE-cadherin and is involved the internalization and ubiquitination of VE-cadherin.”

Response: We thank the reviewer for the comments.

“Lots of the data are gathered from in vivo experiments and show solid results. Some general remarks: to avoid any potential side effects, I would recommend adding a second siRNA in fig 1. Include another siRNA for CHFR and measure VE-cad expression (get another siRNA for hCHFR).”

Response: As suggested, we used a second siRNA to deplete CHFR expression in HLMVECs and found that CHFR depletion prevented LPS-induced degradation of VE-cadherin (Supplementary Fig. 1c).

Also, to test if CHFR is specifically targeting VE, I would recommend repeating some of the functional assays where CHFR is reduced in protein in a VE-cad-deficient background, i.e., knock down of VE. These control experiments are important to strengthen the message and point out specificity.

Response: We appreciate reviewer’s suggestion. It is important to define the specificity of CHFR in endothelial cells. Since VE-cadherin expression at endothelial cell-cell junctions is vital for endothelial barrier integrity, we focused on the role of CHFR in mediating VE-cadherin degradation and its consequences on endothelial barrier integrity. We included several lines of evidence supporting the central role of CHFR-mediated VE-cadherin ubiquitylation in mediating endothelial barrier breakdown. Importantly, VE-cadherin depletion can disrupt endothelial cell-cell junctional integrity; therefore, functional assays are not likely reveal something of significance. Thus, the above suggested study for functional assays we don’t think would be helpful.

“1. Ext data fig 1: add kDa along ubi blot and VE and CHFR co-localize, not associate. That conclusion cannot be drawn from IFs. Please add some arrows for clarity in the magnification.”

Response: We now include MW markers “kDa” in all immunoblots including Supplementary Fig 1 (new Figure # 1e-h, Supplementary Fig. 1a-c, Figure 2c, Figure 3g), VE-cadherin and CHFR co-localization (Figure # 2g) and added arrows to indicate the co-localization of CHFR and VE-cadherin (Figure # 2g).

“2. When VE expression is down upon LPS treatment, is also beta-cat/alpha-cat down or is Ve replaced by N-cadherin?”

Response: We now include data showing that LPS-induced downregulation of VE-cadherin interacting proteins VE-PTP and β -catenin in endothelial cells (Supplementary Fig. 1b). Interestingly, LPS had no significant influence on the p120-catenin (Supplementary Fig. 1b). We did not measure N-cadherin expression.

“3. The authors state, “...E3 ligase CHFR primarily attaches K48-linked poly-Ub chains to VE-cadherin...”. Can the authors provide information that reveals where on the intracellular tail of VE the Ubs are attached? Is that near the beta-catenin or p120 catenin binding sites?”

Response: In the revision we include model (Figure # 2l) to show the VE-cadherin ubiquitylation sites.

“4. Data shown in fig 2d are interesting: VE-PTP is also increased. Is VE-PTP also regulated by CHFR and is this needed to start VE-internalization. Literature shows that p120 regulates VE internalization. How does that integrate with the VE-Ubi and degradation in this study. Is VE-p120 interaction increased upon CHFR knockdown?”

Response: Reviewer #1 also had the same concern. In the revision, we showed that expression of p120-catenin and β -catenin was also increased in addition to VE-PTP in *Chfr*^{ΔEC} mice as compared with WT (Figure # 3e). Further, we showed that LPS challenge caused the loss of VE-PTP and β -catenin expression in control endothelial cells, but not in CHFR knockdown cells (Supplementary Fig. 1b). As discussed above, LPS had no significant effect on the expression of p120-catenin in control and CHFR knockdown endothelial cells (Supplementary Fig. 1b). Thus, these results suggest that LPS induces the dissociation of p120-catenin from VE-cadherin.

“5. It is clear that the depletion of CHFR increases the levels of VE-cad. However, it is not clear if these levels are indeed increased on the cell surface. Only showing I/s with VE staining is not enough. To show this properly, they need to do an antibody-feeding experiment.”

Response: Many studies have shown that VE-cadherin is expressed only at endothelial cell-cell junctions in intact microvessels and *in vitro* endothelial cell cultures. Our immunostaining data using control and CHFR knockdown endothelial cells showed that VE-cadherin was predominantly expressed at cell junctions (Figure # 2d). This observation is further supported by the findings that the inflammatory mediator-induced endothelial barrier breakdown was markedly reduced in CHFR knockdown endothelial cells (Figure #2e, f) as well as in *Chfr*^{ΔEC} mice (Figure # 4d, g, Supplementary Fig. 3a-c). These findings together support the claim that augmented VE-cadherin expression strengthened endothelial barrier in association with its binding partners (VE-PTP, p120-catenin and β -catenin); thus, the antibody feeding experiment suggested is not required to support the conclusions of our findings.

“6. For the inflammation data, they need to address the expression levels of at least some of the essential molecules, like selectins and ICAM-1 to really attribute the effects to CHFR-mediated increase in VE expression.”

Response: Reviewer #1 also raised this issue. In the revision, we have included data showing that the expression of PMN adhesive molecules was not altered in endothelial cells of *Chfr*^{ΔEC} mice compared with WT counterparts (Supplementary Fig. 4a, b). These results suggest that CHFR-mediated

increase in VE-cadherin expression has no influence on the TLR4-induced signaling in endothelial cells. Thus, suppression of vascular inflammation in *Chfr*^{ΔEC} mice is due to the stability of VE-cadherin at AJs.

Reviewer#4:

“In this manuscript, Tirupathi and colleagues have identified CHFR as an ubiquitin E3 ligase that targets VE-cadherin for ubiquitination and degradation. The authors have used AVM BIOMED Snapshot Proteomics Microarray system and have identified CHFR as a VE-cadherin binding protein. The authors have generated an EC-specific CHFR KO mice and show that EC deficient CHFR mice are resistant for TLR4-induced VE-cadherin degradation. Although, this paper is an interesting work, however several conceptual and technical issues reduced my enthusiasm for the manuscript.”

Response: In the revised manuscript, we have addressed all of these conceptual and technical issues.

“First, the authors have used AVM BIOMED Snapshot Proteomics Microarray system to identify CHFR as a VE-cadherin binding protein. The authors provide a scant description about this assay. I even could not find any information about this array in the company’s website. From the data provided, I am not clear what regulates the binding of CHFR with VE-cadherin. Additional binding experiments using recombinantly expressed proteins are needed.”

Response: In the revision, we extensively describe AVM BIOMED Snapshot Proteomics Microarray experimental procedure in Methods section. As suggested, we performed additional experiments using recombinantly expressed proteins and showed that CHFR binds to VE-cadherin (Figure # 1f).

“In vitro ubiquitination assay is required to establish the direct role of CHFR in the ubiquitination of VE-cadherin.”

Response: As suggested, we performed *in vitro* ubiquitylation assay. Results clearly show that CHFR induces K⁴⁸-linked ubiquitylation of VE-cadherin (Figure # 1g, h).

“Second, CHFR is not known to be present at the plasma membrane where the VE-cadherin is present (CHFR is known to present in the nucleus) this raises a serious question whether CHFR is a physiological regulator of VE-cadherin. Extensive co-localization experiments are needed to address this issue.”

Response: In this study, we aimed to define whether the E3 ligase CHFR mediates ubiquitylation of lysine residues present in the cytosolic region of VE-cadherin. Most published studies have used cancer cell lines and found that CHFR primarily in the nucleus. There currently virtually nothing is known about the expression and function of CHFR in endothelial cells. In the revised manuscript, we include confocal imaging data, which demonstrate in endothelial cells that CHFR is present in the cytosol and nucleus (Figure # 2g). Importantly, CHFR is co-localized with VE-cadherin at cell-cell junction (Figure # 2g). Upon LPS challenge, CHFR is also co-localized with VE-cadherin in the endocytosed vesicles (Figure # 2g). These results strongly support our contention that CHFR-mediated ubiquitylation of lysine residues in the cytosolic region of VE-cadherin is a requisite for endothelial barrier breakdown.

“Third, inactivation of VE-cadherin in mice is lethal with severe impairment in angiogenesis, what is the effect of loss of CHFR in angiogenesis.”

Response: We addressed this concern above in response to Reviewer #1. We also include additional experimental results which show that CHFR deficient endothelial cells proliferated at faster rate than control endothelial cells (Supplementary Fig. 5a-c).

“In addition, the authors need to carry out both in vivo and in vitro vascular permeability assays.”

Response: Additional *in vitro* and *in vivo* vascular permeability measurements are provided in the revision (Figures # 2e-f; 4d, g; Supplementary Fig. 3a-c).

Reviewer #5:

“In this study, Tirupathi et al use a protein microarray to scout for binding partners of VE-cadherin. The goal of this was to identify ubiquitin ligases that regulate VE-cadherin levels. CHFR was identified as one of the hits and biochemical and cellular validation experiments were carried out to demonstrate that CHFR mediates K48-linked ubiquitination and consecutive degradation of VE-cadherin. Experiments in primary human lung microvascular endothelial cells revealed that LPS induced degradation of VE-cadherin was prevented by siRNA mediated knockdown of CHFR. In the second part of the manuscript, the authors show that in lung endothelial cells of CHFR knockout in mice VE-cadherin levels are increased and this suppresses LPS-induced vascular injury and mortality. Finally, the authors identify FoxO1 as the responsible transcription factor driving the expression of CHFR after TLR4 agonist stimulation.”

Response: We thank the reviewer for carefully reviewing our manuscript.

“The manuscript is written clearly but needs revision for language in some parts.”

Response: We have fixed this issue in the revision.

“The plots in the figures are not always informative and controls are missing for many experiments (see my comments below).”

Response: As suggested, the figures are now more informative and controls are clearly described in the revision.

“I also find small inconsistencies in some experiments. The methods used need to be described in much more detail (e.g., IP buffer recipe not described, washing of the IPs, affinity purification of peptides for LC-MS etc.).”

Response: We have included expanded LC-MS experimental protocol in the Methods section.

“Comments: AVM BIOMED Snapshot Proteomics Microarray system: explain in more detail (e.g., what negative control was used in the assay and how was the significance assessed?).”

Response: In the revision, we have provided more details about AVM BIOMED Snapshot Proteomics in Methods. Also, we have indicated the negative control used.

“It is not clear to me why the authors decided to focus on CHFR? It has the highest fold-change in the microarray but also the highest p-value among the 20 proteins shown in Extended Data Figure 1b. The same is true for E3 ligases shown in Extended Data Figure 1c. The authors should present a global view of the screening data, for example in form of a volcano plot, and include it in the main figure.”

Response: In the revision, we show the global view of the screening data in the form of volcano and scattered plots as main Figure 1b and 1c. Since CHFR showed highest binding mean value with significant p-value, we focused on the role of CHFR in endothelial cells.

“Extended Data Figure 1f: Are the expression levels of GFP-CHFR (wt, ΔFHA, ΔRF) comparable? A control blot for CHFR is missing.”

Response: Reviewer #4 also had concern about the extended data Figure 1. Therefore, we repeated this experiment and results are shown in Figure #1e, f. We made several deletion CHFR mutant constructs and ectopically expressed them in endothelial cell line HMEC. We observed that ΔRF-CHFR expression was increased compared with WT-CHFR because of the lack of autoubiquitylation and the ability to degrade (Ref # 28). We also observed that reduced expression of ΔFHA-CHFR, ΔPBZ-CHFR as compared with WT-CHFR (Figure # 1e). These results are discussed in the revised manuscript. As suggested, in the revision we have now included control blots (Figure #1f, g, h).

“As additional control, could the authors show that any other of the top-scoring E3s shown in ED Figure 1c does not induce K48-linked polyubiquitylation of VE-cadherin?”

Response: We appreciate reviewer’s suggestion. The other E3s interacting with VE-cadherin were ranked at the bottom of the list and exhibited several-fold decreased binding to VE-cadherin compared with CHFR (Figure # 1c). Therefore, in the present studies we focused on the function of CHFR in endothelial cells.

“Extended Data Figure 1g: Does not look very convincing to me. Would benefit from a proper quantification.”

Response: We performed new experiments in endothelial cells to improve the image quality and quantify the images (Figure # 2g).

“Figure 1b: Show WB for CHFR to demonstrate efficient knockdown. Since TLR4 stimulation results in upregulation of CHFR transcription (through FoxO1), I would expect to see a stabilization of CHFR in this experiment (at least at 24 h of stimulation). In general, all experiments that assess either knockdown/knockout of CHFR or inhibition/depletion of FoxO1, with or without LPS stimulation should include blots for VE-cadherin and CHFR.”

Response: We now include Western blots for CHFR and VE-cadherin all experiments that assess either knockdown/knockout of CHFR or inhibition/depletion of FoxO1, with or without LPS (Figures # 2b, 2c, 3g, 6f).

“Figure 1e: Could the authors comment on the fact that MG-132 treatment does not rescue LPS-induced degradation of VE-cad (see immunoblot for VE-Cad)?”

Response: It is possible that concentration and duration of MG-132 treatment were not sufficient to effectively block the degradation of VE-cadherin by the 26S proteasome complex.

“On top of showing the immunoblot for VE-Cad, I would also recommend to re-probe for VE-cad using the membrane that was utilized for K48 ubiquitin detection.”

Response: As suggested we re-probed K⁴⁸-ubiquitin blot with VE-cadherin (Figures # 1h; 2c; 3g).

“Also, the control blot showing efficient CHFR knockdown is missing. The same is true for figures 2f and g.”

Response: In the revision, we have now included control blot showing efficient CHFR knockdown (new Figure #2b). Also, we included CHFR blot in Figure 2f and g (New Figure # 3g).

“The authors immunoprecipitate VE-cad, followed by enrichment of diGly remnant peptides to investigate whether LPS stimulation induced VE-cad ubiquitination. Some more details on the setup should be described in the figure legend. For example, that the experiment was done in the presence of MG-132 and that the treatment time was 6 h.”

Response: In the revision, we included expanded experimental details in the figure legend as well in Methods.

“I am wondering why the authors chose a double IP approach (i.e., IP VE-cad + IP diGly), instead of directly enriching for K-GG peptides after LPS treatment?”

Response: We focused on the mapping of “K” residues ubiquitylated in the C-terminus of VE-cadherin in endothelial cells. LPS/TLR4 axis activates many downstream signaling pathways, which primarily depends on ubiquitylation and deubiquitylation of multiple signaling components. Thus, the total cell lysate digestion with trypsin could result in excessive production of K-GG peptides from many proteins which could exceed the threshold for K-GG peptide antibody. It may also be that the protein of interest K-GG peptide concentration is a small fraction of the total peptides; this could pose a technical problem in detecting the VE-cadherin-derived peptides by LC/MS. To overcome this issue, we immunoprecipitated (IP-ed) the protein of interest VE-cadherin and then used the IP-ed proteins for LC/MS analysis.

“Instead of showing MS/MS spectra in the main figure (which are not very informative), the authors should present quantitative results (i.e., levels of all quantified K-GG peptides in control vs treatment vs the corresponding levels of total protein).”

Response: We performed qualitative LC-MS to uncover the “K” residues ubiquitylated in the C-terminus of VE-cadherin. We identified two ubiquitylated peptides (Figure # 2i-j) and quantified the total ion chromatograms (TIC) of these two peptides. Here, we observed that substantially increased amounts of VE-cadherin-derived ubiquitylated peptides in LPS treated samples compared control (Figures # 2i, j).

“The authors claim that CHFR attaches K48-linked ubiquitin to K633 and K690 of VE-cad; however, diGly remnant profiling does not allow determining the ubiquitin linkage type. This statement should be removed or else an orthogonal, MS-based approach (e.g., TUBEs (tandem ubiquitin binding entities)) should be employed.”

Response: We appreciate reviewer’s suggestion. Our findings are in-agreement with the concept that in ubiquitin-proteasome system, proteins modified by K48-linked polyubiquitin chains are the “signature” for the 26S proteasome-mediated degradation. We used validated ubiquitin linkage specific antibodies to identify whether CHFR attaches either K48- or K63-linked polyubiquitin chains to the “K” residues of VE-cadherin. In many experiments including *in vitro* ubiquitylation assay we showed that CHFR induces K48-linked polyubiquitylation of VE-cadherin (Figures # 1h; 2c; 3g; 6g). Therefore, our results point to CHFR attachment of K48-linked polyubiquitin chains to VE-cadherin.

“A negative control should be included for the experiment shown in figure 1i (for example, VE-pep-1 in which the lysine is substituted by an arginine).”

Response: As suggested, we synthesized peptides where lysine is substituted by arginine (i.e., K/R) for negative control. Here, we observed that the negative control peptides had no effect on LPS-induced degradation of VE-cadherin from the cell junctions (Supplementary Fig. 1d).

REVIEWER COMMENTS

Reviewer #1 (Remarks to the Author):

As the authors responded, the study is focused on the role of CHFR in mediating the ubiquitylation of VE-cadherin and its effect on endothelial barrier integrity. However, this reviewer feels that the sole credit for the mechanisms is given to EC VE-Cadherin. A bulk RNAseq or scRNAseq analysis should help to better interpret the data, particularly because this is a new mouse model. At least some speculations should be discussed with supporting literature if the RNAseq is out of scope.

Reviewer #2 (Remarks to the Author):

The authors have responded to the reviewer's comments by several new experiments and including new results.

There is one important issue left which is not yet adequately addressed:

1) Whereas I appreciate the efforts to repeat the co-localization data for CHFR and VE-cadherin, I am not convinced that the images shown in Fig. 2g are meaningful. Especially the "co-staining" in the nucleus does not make much sense as sites of the endocytic compartment. The staining pattern for CHFR is so broad that any intracellular staining of VE-cadherin (specific or not) could be superimposed. The structures that show co-localization are ill defined.

Reviewer #3 (Remarks to the Author):

The authors have thoroughly responded to my comments and answered all my concerns to my satisfaction.

Reviewer #5 (Remarks to the Author):

The authors answered most of my concern adequately. I do, however, disagree with parts of the response regarding figure 2i/j and my suggestion would be to remove figure panels 2 i/j/k from the manuscript. Overall, they add little information to the manuscript and both the experimental setup as well the data analysis would need major revision.

Here is my justification:

According to the description in the methods section, the authors incubated the cells with LPS (5 µg/ml) for 6 h in the presence of 10 µM MG-132 and compared levels of the VE-cad diGly peptides before and after stimulation (i.e., with or without MG132 addition). However, in this way, the effect of MG-132 cannot be distinguished from the LPS effect since the ubiquitinome signal would increase globally. The correct way of doing this is to compare MG-132 treated cells +/- LPS stimulation (I guess a much shorter stimulation would also work).

Moreover, the authors quantify the K633/K690 diGly peptides by integrating the total ion chromatogram, but they probably meant the base peak intensity (and present MS/MS spectra in the figure)?? The differences they show are really small (20-30% increase only). On top, the experiment was performed only once and therefore no statistical significance can be assessed. A (statistically significant) 20-30% fold-change with label-free quantification is very difficult to detect, even for regular proteomics. In this case, the authors have used a tandem purification strategy, which introduces a lot of variances and on top the experiment was performed just once and it is not clear whether and how the data was normalized. The presented data is not convincing at all (and the setup of the experiment is not correct) and I would thus suggest removing the figure from the manuscript or else repeat the experiment and assess statistical significance correctly (as well as changing the setup, see above).

I appreciate that K48-linked ubiquitination can trigger proteasomal degradation and that the authors demonstrate K48 linkages which are mediated by CHFR. However, I do not find any biochemical proof that the identified di-Gly peptides derive from K48-linked ubiquitin. Therefore, I would insist to remove the claim that CHFR attaches K48-linked ubiquitin to residues K633 and K690 of VE-cad.

Response to reviewers' comments:

Reviewer#1:

“As the authors responded, the study is focused on the role of CHFR in mediating the ubiquitylation of VE-cadherin and its effect on endothelial barrier integrity. However, this reviewer feels that the sole credit for the mechanisms is given to EC VE-Cadherin. A bulk RNAseq or scRNAseq analysis should help to better interpret the data, particularly because this is a new mouse model. At least some speculations should be discussed with supporting literature if the RNAseq is out of scope.”

Response: We greatly appreciate the reviewer's suggestion. In response, we now speculate on the value of the new mouse model (*Chfr^{AEC}*) and how the studies have helped to illuminate the control of VE-cadherin junctions in endothelial cells. We have also added a reference (Morini et al., *Circ Res.* 2018; 122:231-245) to provide a context to our work. Finally, we have discussed in detail the novel concept emerging from our studies that VE-cadherin itself upregulates endothelial genes known to contribute to vascular stability through inhibiting polycomb repressive complex-2 (PRC2) function in endothelial cells (Page #13).

Reviewer #2:

“The authors have responded to the reviewer's comments by several new experiments and including new results. There is one important issue left which is not yet adequately addressed: 1) Whereas I appreciate the efforts to repeat the co-localization data for CHFR and VE-cadherin, I am not convinced that the images shown in Fig. 2g are meaningful. Especially the “co-staining” in the nucleus does not make much sense as sites of the endocytic compartment. The staining pattern for CHFR is so broad that any intracellular staining of VE-cadherin (specific or not) could be superimposed. The structures that show co-localization are ill defined.”

Response: In response, we repeated the confocal experiments. The revised manuscript shows images (Figure 2g) that clearly show co-localization of VE-cadherin with CHFR in the endocytic compartment located at the junctions.

Reviewer #3:

“The authors have thoroughly responded to my comments and answered all my concerns to my satisfaction.”

Response: Thank you.

Reviewer #5:

“The authors answered most of my concern adequately. I do, however, disagree with parts of the response regarding figure 2i/j and my suggestion would be to remove figure panels 2 i/j/k from the manuscript. Overall, they add little information to the manuscript and both the experimental setup as well as the data analysis would need major revision.”

Response: As suggested, we have deleted the LC-MS results panels in Figure 2 and revised both the experimental setup and data analysis sections. We agree this aspect of the paper detracts from the work.

“Here is my justification: According to the description in the methods section, the authors incubated the cells with LPS (5 µg/ml) for 6 h in the presence of 10 µM MG-132 and compared levels of the VE-cad diGly peptides before and after stimulation (i.e., with or without MG132 addition). However, in this way, the effect of MG-132 cannot be distinguished from the LPS effect since the ubiquitinome signal would increase globally. The correct way of doing this is to compare MG-132 treated cells +/- LPS stimulation (I guess a much shorter stimulation would also work). Moreover, the authors quantify the K633/K690 diGly peptides by integrating the total ion chromatogram, but they probably meant the base peak intensity (and present MS/MS spectra in the figure)?? The differences they show are really small (20-30% increase only). On top, the experiment was performed only once and therefore no statistical significance can be assessed. A (statistically significant) 20-30% fold-change with label-free quantification is very difficult to detect, even for regular proteomics. In this case, the authors have used a tandem purification strategy, which introduces a lot of variances and on top the experiment was performed just once and it is not clear whether and how the data was normalized. The presented data is not convincing at all (and the setup of the experiment is not correct) and I would thus suggest removing the figure from the manuscript or else repeat the experiment and assess statistical significance correctly (as well as changing the setup, see above).”

Response: We appreciate reviewer’s suggestions. These will be very useful for our future experiments.

“I appreciate that K48-linked ubiquitination can trigger proteasomal degradation and that the authors demonstrate K48 linkages which are mediated by CHFR. However, I do not find any biochemical proof that the identified di-Gly peptides derive from K48-linked ubiquitin. Therefore, I would insist to remove the claim that CHFR attaches K48-linked ubiquitin to residues K633 and K690 of VE-cad.”

Response: In the revision, we have removed the claim that CHFR attaches K48-linked ubiquitin to residues K633 and K690 of VE-cadherin.